



# Estimating surface mass balance patterns from UAV measurements on the ablation area of the Morteratsch-Pers glacier complex (Switzerland)

Lander VAN TRICHT[1,*], Philippe HUYBRECHTS[1], Jonas VAN BREEDAM[1], Alexander VANHULLE[1], Kristof VAN OOST[2], Harry ZEKOLLARI[3]

[1] Earth System Science & Departement Geografie, Vrije Universiteit Brussel, Brussels, Belgium

[2] Georges Lemaître Center for Earth & Climate Research, Earth and Life Institute, Université catholique de Louvain, Louvain-la-Neuve, Belgium

[3] Department of Geoscience and Remote Sensing, Delft University of Technology, Delft, Netherlands

*Corresponding author: Lander Van Tricht (lander.van.tricht@vub.be)

**Abstract.** The surface mass balance of a glacier (SMB) provides the link between the glacier and the local climate. For this reason, it is intensively studied and monitored. However, major efforts are required to determine the SMB on a sufficient number of locations to capture the heterogeneity of the SMB pattern. Furthermore, because of the time-consuming and costly nature of these measurements, detailed SMB measurements are carried out on only a limited number of glaciers. In this study, we investigate how to accurately determine the SMB in the ablation zone of Vadret da Morteratsch and Vadret Pers (Engadin, Switzerland) using the continuity-equation method. For this, an elaborate dataset (spanning the 2017-2020 period) of high-resolution data derived from UAV measurements (surface elevation changes and surface velocities) is combined with reconstructed ice thickness fields (based on radar measurements). To determine the performance of the method, we compare modelled SMB with measured SMB values at the position of stakes. Our results indicate that with annual UAV surveys, it is possible to obtain SMB estimates with a mean absolute error of approximately 0.5 metre ice equivalent per year. Yet, our study demonstrates that in order to obtain these accuracies, it is necessary to consider the ice flow over spatial scales of several times the local ice thickness using an exponential decay filter. Furthermore, our study shows the crucial importance of the ice thickness, which must be sufficiently well known in order to apply the method. The latter currently hampers the application of the continuity-equation method to derive detailed SMB patterns on regional to global scales.



## 1 Introduction

The surface mass balance of a glacier is determined by the processes adding mass to the surface (e.g. snow fall, freezing rain), and those removing mass from the surface (e.g. snow and ice melt, sublimation). These processes are strongly driven by the local temperature and precipitation over the glacier. As a result of increased global mean temperatures, SMBs are becoming increasingly negative, leading to an unprecedented shrinkage of glaciers during the last decade (Zemp et al., 2019; Wouters et al., 2019). Because of the direct link with the local climatic signal, determining the glacier surface mass balance and its distribution is crucial to monitor, understand and model the reaction of glaciers to climate change. Traditionally, a stake and snow pit network is used to determine the SMB at a small number of positions followed by an inter- and extrapolation to obtain the glacier average specific mass balance (Braithwaite, 2002). This can result in large errors for glaciers where the heterogeneity of the SMB cannot be captured sufficiently by the available measurements (Zemp et al., 2013). Further, because of the time-consuming and costly nature of these measurements, detailed SMB measurements are carried out on only a limited number of glaciers.

Geodetic methods provide an alternative, and have lately been applied in numerous studies, to monitor mass balances for individual glaciers at local to regional scales (e.g. Brun et al., 2017; Davaze et al., 2020; Sommer et al., 2020). These methods all involve comparing digital elevation models (DEMs), mainly created using airborne and satellite data, over a given period to determine local elevation changes. These local elevation changes result from an interaction between mass balance and ice flow processes and do therefore not allow to directly determine the mass balance distribution over glaciers. However, having such a mass balance distribution over glaciers is of large interest, as this can be used to accurately calibrate mass balance models used in large-scale glacier modelling efforts to allow for example for an accurate calibration of mass balance gradients (Zekollari et al., 2018; Marzeion et al., 2020).

Several studies have attempted to determine patterns of SMB from surface elevation changes by implementing either mass-continuity, a kinematic boundary condition at the surface or through 3D ice flow modelling (Kääb and Funk, 1999; Gudmundsson and Bauder, 1999; Hubbard et al., 2000; Reeh et al., 2002; Nuimura et al., 2011; Vincent et al., 2016; Bisset et al., 2020). In essence, these are all based on the principle of combining surface elevation changes with the ice flux divergence. While the former can be measured directly, the latter is calculated by combining various types of measurements. Most of previous studies stumbled on the necessity, resulting from the discretization of ice flow processes and spatial resolution of the data, to smooth the input data or the ice flux divergence or to resample to much lower resolutions of several hundreds of metres. Further, a recurring drawback in previous studies attempting to detect local and small-scale variations of the SMB was the lack of satellite observations with sufficient spatial and temporal resolution (Ryan et al., 2015). However, with the emergence of Unmanned Aerial Vehicles (UAV), it turned out to be possible to detect small scale variations at unprecedented centimetre resolution. In recent years, studies have attempted to determine the SMB at the location of individual ablation stakes using an UAV and with measured vertical velocities (e.g. Vincent et al., 2020). However, to our knowledge, no research has yet been carried out to derive a transferable method to determine the SMB distribution over an entire ablation zone using multiannual UAV measurements. Furthermore, the optimal ways to calculate



the ice flux divergence, and how to represent the spatial scales over which ice flow occurs without resampling to
a much lower resolution, remain a topic of discussion.
The aim of this study consists of deriving and applying a method to determine the SMB in the entire ablation zone
of two glaciers by combining UAV acquired 3D data and ice thickness measurements through the continuity-
equation method. We pay particular attention to the spatial scales that need to be considered in this framework and
how these influence the modelled SMB. The performance is evaluated through an in-depth comparison with
measured SMB at stakes. To allow the method to be applied for other glaciers, with different data availability, we
also perform a comprehensive sensitivity analysis, from which we determine which data is crucial in the
application of the method and its accuracy.

## 2 Study area and fieldwork

The current study focuses on the Morteratsch - Pers glacier complex situated in the Bernina massif in the
southeastern part of Switzerland (European Alps) (Figure 1). Vadret da Morteratsch, with a current length of
approximately 6 km, is the main glacier and flows from the south to the north. Vadret Pers, which flows towards
Vadret da Morteratsch from the southeast, became a separate glacier in 2015, when both glaciers disconnected
(Zekollari and Huybrechts, 2018). At present, the two glaciers cover an area of about 15 km$^2$ with a volume of ca.
1 km$^3$ making it the largest continuous ice area in the Bernina region. Vadret da Morteratsch reached its maximum
Little Ice Age (LIA) extent between 1860 and 1865 (Zekollari et al., 2014). Since 1878, the glacier front has
retreated more than three kilometres and is nowadays located at an altitude of 2200 m above sea level (a.s.l). The
upper parts of the glacier complex reach 4000 m a.s.l, originating at peaks such as Piz Bernina and Bellavista
(Figure 1). The glacier complex has been studied and monitored intensively with SMB stake measurements
performed annually at the end of the ablation season since 2001 (Zekollari and Huybrechts, 2018). In addition, the
geodetic mass balance of the 1980-2010 period was determined to be between -0.7 and -0.8 metre water equivalent
per year (Fischer et al., 2015). The SMB stake measurements served to calibrate a mass balance model (Nemec et
al., 2009) which was coupled to a higher-order ice flow model to simulate the future evolution of the glacier
complex (Zekollari et al., 2014) and to study its response time (Zekollari and Huybrechts, 2015). Furthermore, the
ice thickness in the ablation area has been measured twice (Zekollari et al., 2013; Langhammer et al., 2019) and
again in 2020 for specific parts of Vadret da Morteratsch.



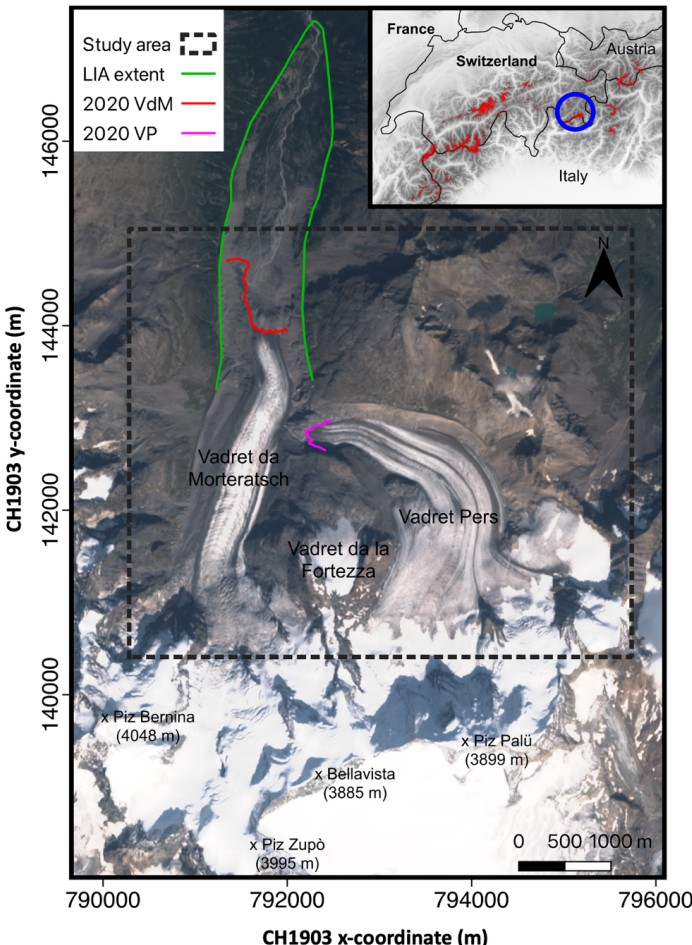

**Figure 1**. *Map of the Morteratsch-Pers glacier complex in southeastern Switzerland. The different coloured lines represent the extent of the glacier at the end of the Little Ice Age (LIA, 1860-1865) and in 2020. The background image is a Sentinel-2 true colour composite satellite image from 13 September 2020. The highest mountains are labelled and indicated with a black cross. The inset shows a DEM of the Shuttle Radar Topography Mission (SRTM) and boundaries of the countries of the central and western European Alps. The areas indicated with a red colour are glaciers according to the Randolph Glacier Inventory (RGI) version 6. The location of the Morteratsch-Pers glacier complex is indicated with a blue circle.*

## 2.1 UAV surveys

Between 2017 and 2020, we conducted annual UAV surveys at the end of the ablation season (late September). Hence, the focus in this research is on three mass-balance years (2017-2018, 2018-2019 and 2019-2020). In 2017, above normal snow cover prevented drone mapping on the upper parts of the ablation area of Vadret Pers (>2800



m) due to inadequate visual content for accurate keypoint detection (Gindraux et al., 2017). This resulted in a
smaller study area in 2017-2018. The observations consist of images acquired by repeated UAV surveys with a
DJI Phantom 4 Pro (P4PRO) (in 2017, 2018 and 2019) and a DJI Phantom 4 RTK (P4RTK) (in 2020) quadcopter,
both equipped with a 20-megapixel camera. For flight planning and UAV piloting, DJI GS Pro was used. The
flight plans were designed to limit the variation in ground sampling distance (GSD) by flying parallel to the main
surface slope and by subdividing the study area into smaller sections. Additional technical details of the flights are
given in Table 1. The different flights within one field campaign were performed on multiple days. However,
because of the limited surface change during the fieldwork periods (4-6 days), no correction was applied for the
difference in acquisition dates.


***Table 1.*** *Technical details of the UAV flights.*

| Setting | Value |
|---|---|
| **Flight altitude** | On average 180 m |
| **Ground resolution** | 5-10 cm |
| **View angle** | 90° |
| **Flight speed** | 5 m/s |
| **Capture interval** | 4 s |
| **Frontal overlap** | 90% |
| **Side overlap** | 70% |



To ensure sufficient horizontal and vertical accuracy, ground control points (GCPs) were distributed over the area
of interest. The GCPs, plastic orange squares of 40x40 cm, were measured with a Trimble 7 GeoXH RTK GPS
(horizontal average accuracy of 10-20 cm, vertical average accuracy of 20-30 cm) by relying on the swipos
positioning service. The GCPs were spread over different locations on the glacier following density and
distribution guidelines from the literature (Tahar et al., 2012; Goldstein et al., 2015; Long et al., 2016; Tonkin et
al., 2016; Gindraux et al., 2017). We ensured homogeneous distribution in almost every case, except where
crevasses, moulins or a fresh layer of snow (especially in 2017) impeded this with an average density of 10-20
GCPs km$^{-2}$. In 2020, a smaller number of GCPs was placed because the P4RTK can achieve centimetre-level
accuracy without a large number of GCPs as a result of an on-board differential GPS system (Zhang et al., 2019;
Kienholz et al., 2020).


**2.2  In situ surface mass balance data**

A total of 287 annual mass balance stake measurements (158 on Morteratsch, 129 on Pers) were performed
between 2001 and 2020 in the ablation area of the glacier complex. All the stakes were located between 2100 and
2600 m a.s.l. on the Morteratsch glacier's ablation tongue and between 2450 and 3050 (approximately the ELA)
on Vadret Pers. Zekollari and Huybrechts (2018) summarized these measurements and found the SMB on Pers
glacier to be significantly lower (2.1-2.5 metres ice equivalent (m i.e.) yr$^{-1}$) compared to Vadret da Morteratsch at





similar elevation. This has been mainly attributed to differences in orientation and the associated daily insolation
cycle over the two glaciers. For the period of concern in this study (2017-2020), SMB measurements from 16
individual stakes (8 on each glacier) are available for every year on the debris-free part of the ablation areas (Figure
2). The stakes were measured, and replaced if necessary, annually at the end of each ablation season, during the
UAV surveys.


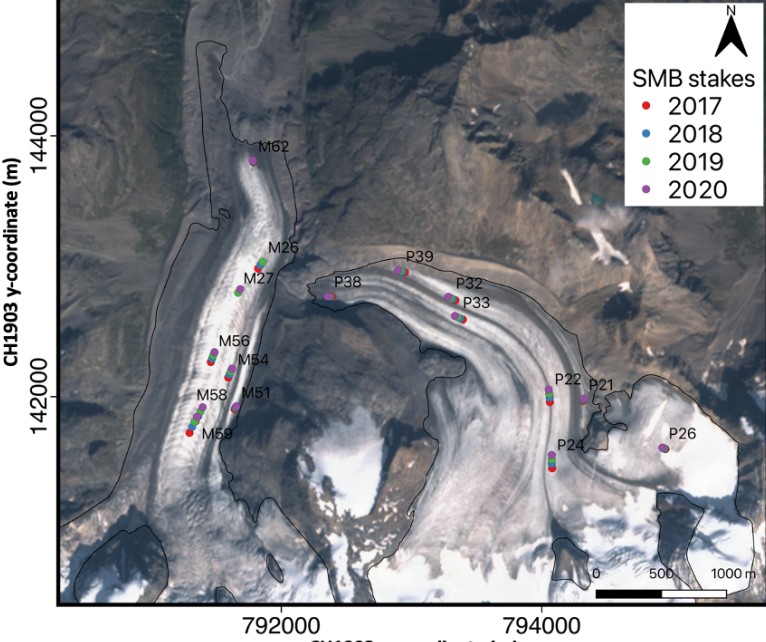



**Figure 2**. *Satellite image of the study area with the location of the measured ablation stakes in 2017, 2018, 2019 and 2020.*
*The background image is a Sentinel-2 true colour composite satellite image from 13 September 2020. The different locations*
*of the stakes in 2017-2018-2019-2020 show the movement of the stakes with the glacier flow. The labels refer to the stake*
*names as used in the fieldwork programme. In 2019, stake M26, which was located in the middle of a crevasse field, was*
*replaced by stake M27.*


**2.3  Ice thickness data**

To calculate the local ice volume flux divergence (see section 3.4), a distribution of the ice thickness is required.
There are two published datasets of the ice thickness of the Morteratsch – Pers glacier complex, derived from radar
measurements (especially in the ablation area) and modelling (especially in the accumulation area). The first one
is the dataset of Zekollari et al. (2013), hereafter referred to as THIZ. This distribution was reconstructed by
combining measured ice thicknesses (~30 ground borne GPR profiles in the early 2000s) with different modelling



constraints. The second ice thickness distribution is the dataset of Langhammer et al. (2019), hereafter referred to as THIL. This distribution was produced by combining measured ice thicknesses (41 helicopter borne GPR profiles with a total of 53247 points measured in 2017) and modelling constraints with the Glacier Thickness Estimation (GlaTE) inversion scheme. We correct both datasets for glacier geometry changes to refer to the glacier state in 2018, 2019 and 2020 respectively by subtracting the amount of local surface lowering. This concerns the local height difference between the UAV created DSMs of 2018-2019-2020 and the DEM used for the respective ice thickness datasets. The latter are two DEMs provided by SwissTopo. DHM25 valid for 1991 (used for THIZ) and SwissALTI3D valid for 2015 (used for THIL).

Although both ice thickness distributions (Figure 3a and Figure 3b) have a similar pattern (location of overdeepenings, location of maximum ice thickness), large local differences exist. The ice thickness maximum for instance occurs at a similar location but is about 310 m for THIZ and 250 m for THIL. This corresponds to a difference of 15-20% which is however still within the error bounds of the datasets. Conversely, there are also certain zones (especially higher up Vadret da Morteratsch) where the difference between the two datasets is more than 150 metres.

Because of the significant differences between both ice thickness distributions, we hypothesize that the choice for a particular distribution can have a major influence for the calculations in this study. To verify the maximum ice thickness of Vadret da Morteratsch, we measured the ice thickness once again at the thickest point in 2020 with a Narod Radio Eco Sounding (RES) system, similar to Van Tricht et al. (2020). We preferred to use a low frequency of 5 MHz to limit the amount of attenuation due to disturbances in the glacier such as water inclusions, repeatedly observed during the fieldwork. We found a maximum value of 296 metres which is relatively close to the maximum thickness from the THIZ dataset. We can therefore certainly not ignore this dataset, despite the smaller number of measurements and the older date of creation. Therefore, for further calculations, we decided to use the mean ice thickness from THIZ and THIL at every grid cell, hereafter referred to as THIA (Figure 3c). The effect of using THIA, as opposed to relying on THIZ or THIL, is examined as a part of the sensitivity analysis (section 5.3). For the areas where the current elevation is lower than the bedrock inferred from THIZ and THIL, and where there is no ice as a result, we assume a minimal ice thickness of 5m as in Zekollari et al. (2013).





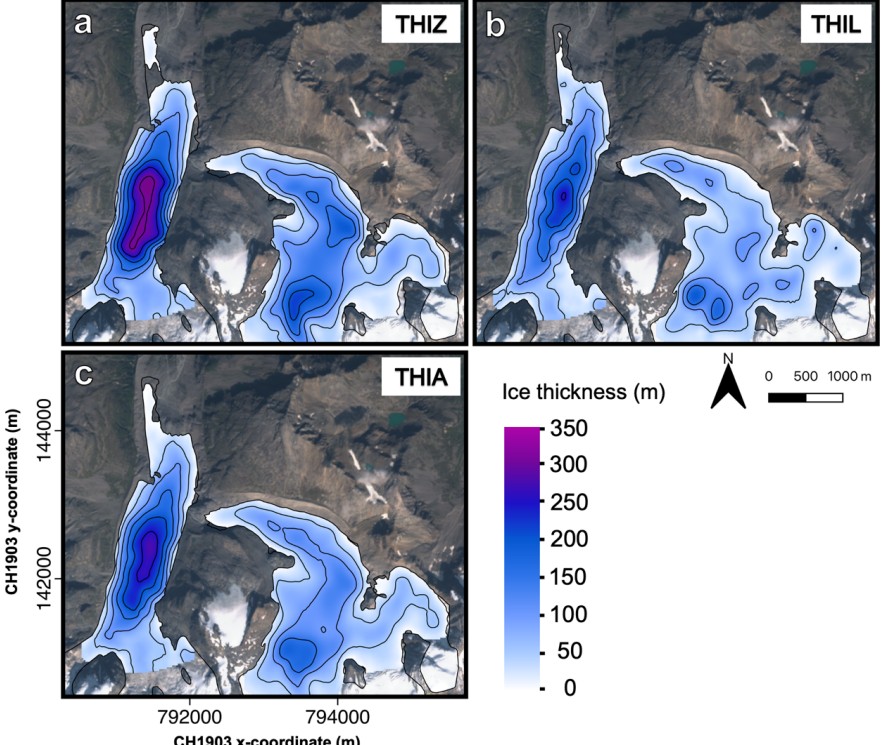

**Figure 3.** Ice thickness distribution of the ablation area of the Morteratsch-Pers glacier complex in 2020. (a) THIZ (ice thickness distribution of Zekollari et al. (2013), (b) THIL (ice thickness distribution of Langhammer et al. (2019) and (c) THIA (average ice thickness distribution). Contour lines represent 50 m intervals. Note that at the front of Vadret da Morteratsch and locally at the front of Vadret Pers, the glacier outline is larger than the ice thickness dataset. The current surface elevation of these areas is lower than the bedrock elevation inferred from THIL and THIZ. The background image is a Sentinel-2 true colour composite satellite image from 13 September 2020.

## 3    Methods

### 3.1  Continuity-equation method

The continuity equation for glacier flow with constant density (Eq. 1) links the local mass balance (b) and the ice flux divergence per unit width in the local vertical ice column ($\nabla \cdot \vec{q}$) with local changes in the ice thickness ($\frac{\partial H}{\partial t}$), all expressed in metres of ice equivalent per year (m i.e. yr$^{-1}$) (see e.g. Hubbard et al., 2000; Berthier and Vincent, 2012).



$$\frac{\partial H}{\partial t} = b \ - \ \nabla.\vec{q} \tag{1}$$

Eq. 1 shows that the difference between the local mass balance and ice flux divergence must be compensated by a change in local ice thickness. If the bedrock elevation is assumed to be constant and compaction is negligible, which is the case in the ablation area (with ice throughout the entire column), the local ice thickness (H) in Eq. 1 can be replaced by the elevation of the surface (h). Further, becuase basal and internal mass balance in the ablation area are predominantly much smaller compared to the surface mass balance ($b_s$) (Kaser et al., 2003; Huss et al., 2015), b can be replaced by $b_s$, which after a reorganisation leads to the following expression:

$$b_s = \frac{\partial h}{\partial t} + \ \nabla.\vec{q} \tag{2}$$

By determining the local elevation changes (section 3.2) and the components that make up for the ice flux divergence per unit width (from now on referred to as ice flux divergence; sections 3.3-3.6), the SMB pattern can then be derived.

**3.2 DSM generation and surface elevation changes ($\partial h/\partial t$)**

First, all the data from the UAV surveys are used to generate DSMs on a common local coordinate system (CH1903 LV03). All the images collected during the UAV surveys are processed into orthophotos and DSMs using the photogrammetry workflow implemented in Pix4D. The accuracy of the reconstructed DSMs is assessed using ground validation points (GVPs), which are GCPs that are not used to georeference the project. Subsequently, surface elevation changes are directly computed from these DSMs by subtracting the DSMs from each other (2018-2017, 2019-2018, 2020-2019). Initially, all the DSMs are generated with a very high resolution of 0.05-0.10 m. But eventually, the surface elevation changes are resampled to 25 m resolution for further calculations using a block moving average filter. This corresponds to the resolution of the ice thickness datasets and will therefore be the grid size for all of the following calculations.

A commonly observed feature on surface elevation change maps derived from high resolution DSMs are alternating positive and negative differences. These are caused by the advection of local glacier topography such as crevasses, moulins and large ice or rock boulders (Figure 4) (Rounce et al., 2018; Yang et al., 2020). The above-mentioned variations are not caused by reduced or increased melt or accumulation and therefore need to be filtered out in order to make a correct SMB estimate.

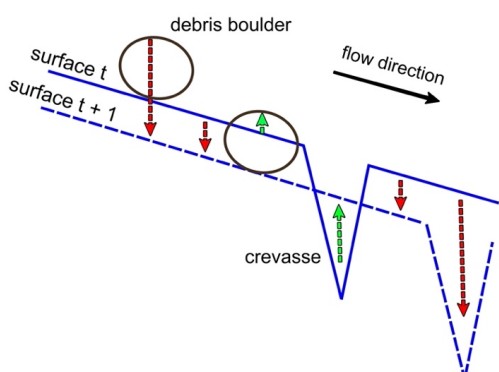



**Figure 4**. *The advection of surface topography (such as debris boulders and crevasses) can alter surface elevation changes*
*between timestep t and t+1 (red = negative ∂h/∂t, green = positive ∂h/∂t). The net effect on the surface elevation depends on*
*the amount of lowering and the vertical extent of the moving features on the glacier surface.*


To filter out these variations, we apply a block moving average filter (Eq. 3) with a window size d (in number of
grid cells) which is the product of a smoothing parameter (k) and the local surface velocity ($u_s$) (Eq. 4).


$$\frac{\partial h}{\partial t}[x,y] = \frac{1}{(2d+1)^2}\left(\sum_{i=-d}^{d}\sum_{j=-d}^{d}\frac{\partial h}{\partial t}[x+i,y+j]\right)$$    (3)

$$d = k * \text{round}\left(\frac{u_s}{25}\right)$$    (4)


x and y are Cartesian coordinates, and i and j are indices. In contrast to a block moving average filter with constant
width, the surface elevation changes are only smoothed in areas where the glacier is flowing. In areas where the
glacier is (almost) stagnant, ∂h/∂t variations are kept as these cannot have been caused by the movement. Hence,
∂h/∂t is not modified when d is smaller than one, implying that the local glacier movement during a year was
smaller than the grid resolution (see Eq. 3 and Eq. 4). In order to investigate the optimal distance over which the
surface elevation changes need to be smoothed, k is varied between 0 (no modification) and 10. Ideally, k should
not be taken too large to avoid eliminating all variations, including SMB induced variations.


**3.3  Horizontal surface velocity**

Glacier surface velocities, which are needed to calculate the ice flux divergence (section 3.4) and to smooth the
surface elevation changes (section 3.2), are computed for the three individual mass balance years. Velocity grids





are derived by applying the Image Geo Rectification and Feature Tracking toolbox (ImGRAFT), an open-source tool in MATLAB (Messerli and Grinsted, 2015). Instead of using visual image data with variable illumination and snow cover, the feature tracking algorithm is run on hillshades (relief DSMs) of the original DSMs resampled to a resolution of 2 m (Messerli and Grinsted, 2015). The latter turned out to improve the correlation success (Rounce et al., 2018).

Velocity maps are computed and corrected for differences in acquisition dates to obtain values in metre per year. Then, a series of filters is applied to the output of the cross-correlation to exclude poorly correlated pixels and those with unrealistic displacements or flow directions (Ruiz et al., 2015). The latter is a common issue for snow covered areas or debris covered areas where no accurate displacements can be detected. As a first filter, all the computed velocity vectors outside the glacierized areas are removed by using the digitized glacier outlines (Heid and Kääb, 2012). Then, a median filter is applied which removes velocity vectors that deviate too much from the surrounding vectors (Heid and Kääb, 2012; Nagy et al., 2019). Finally, we also impose a maximum velocity threshold derived from the modelled velocities in Zekollari et al. (2013), which are constrained with field observations. In other words, we limit the maximum surface velocity at every grid cell to the velocity modelled in Zekollari et al. (2013) + 20%. The latter is a margin to take into account potential errors present in the study of Zekollari et al. (2013). However, after analysis this proved not to be necessary as these high values did not occur. The raw velocity maps containing gaps after filtering are then interpolated using a spline interpolation (Ruiz et al., 2015). As validation, we compare the obtained velocities with measured velocities from stakes.

### 3.4. Ice flux divergence from ice thickness and surface velocities

The ice flux divergence (in m i.e. $yr^{-1}$) corresponds to the local upward or downward flow of ice relative to the glacier surface. It represents the difference between the ice supplied from upstream and lost downstream at a particular position. It is defined to be negative for upward motion (mass supplied to the surface, also referred to as the emergence velocity) and positive for downward motion (mass is removed from the surface, also referred to as the submergence velocity).

Different approaches exist to calculate the ice flux divergence ranging from the use of 3D ice flow models (Seroussi et al., 2011; Vincent et al., 2020) to the use of simple geometric calculations and flux gates (Nuimura et al., 2011; Berthier and Vincent, 2012). The important distinction is the simplicity and the resolution with which they can be calculated. In this study, the ice flux divergence is computed for each grid cell, by combining surface velocities and ice thickness according to Eq. 5:

$$\nabla.\vec{q} = \quad F \left( u_{s,x} \frac{\partial H}{\partial x} + u_{s,y} \frac{\partial H}{\partial y} + H \frac{\partial u_{s,x}}{\partial x} + H \frac{\partial u_{s,y}}{\partial y} \right) \tag{5}$$



The right side of Eq. 5 contains two horizontal ice flux components of which the first corresponds to the ice thickness divergence and the second one to the velocity divergence (Reeh et al., 2003). Both are computed from the relevant derivatives. F is the depth averaged glacier velocity ratio ($\bar{u}/u_s$). For an isothermal glacier, with negligible basal sliding and Glen's flow coefficient n = 3, F is 0.8. However, the Morteratsch-Pers glacier complex is assumed to be a temperate alpine glacier complex which is accompanied by basal sliding. According to Zekollari et al. (2013), interal deformation accounts glacier wide on average for 70% of the flow and basal sliding for the remaining 30%. However, in the ablation area the contribution of internal deformation is most likely even higher. Therefore, for the standard runs in this study, we take an F-value of 0.9. However, the F-value may vary locally (Zekollari et al., 2013) and varying this parameter will therefore be part of the sensitivity analysis.

### 3.4 Spatial scales of the ice flux divergence

In contrast to the surface elevation changes, for which the accuracy is expected to be high, the computed ice flux divergence is subject to larger uncertainties. This uncertainty directly relates to the large uncertainties in the reconstruced ice thickness field. Moreover, a considerable part of the uncertainty also originates from the spatial scales that need to be considered: due to longitudinal stresses, the local ice dynamics do not only depend on the local glacier geometry (an assumption of the Shallow Ice Approximation, see e.g. Hutter and Morland (1984)), but also on the surrounding geometry, typically over scales corresponding to several times the local ice thickness (accounted for in higher-order and Full Stokes models, see e.g. Zekollari et al. (2014), where such a model was applied for this glacier complex). Local gradients of the ice thickness and the velocity are often magnified when they are determined on a numerical grid with a central difference scheme. This is directly reflected in the ice flux divergence pattern. Therefore, to make the ice flux divergence solution independent of the resolution and to take non-local stresses into account, larger scales must be considered (Reeh et al., 2003; Rounce et al., 2018). In studies applying the continuity equation on larger glaciers or ice sheets, it is usually sufficient to resample the ice flux divergence grid to a much lower resolution (Nuimura et al., 2011; Vincent et al., 2016; Rounce et al., 2018). However, as we aim for a high resolution of the SMB determination in this study, we opt to retain the 25 m resolution and to consider larger scales for the calculation of the ice flux divergence.

Different filters have been applied, of which most are constant box filters (e.g. mean, median) with a strong variation in size (up to 10000 metres) (Kääb and Funk, 1999; Seroussi et al., 2011; Rounce et al., 2018). Such filters give equal weights to cells within the box irrespective of their distance from the point in consideration (Kamb and Echelmeyer, 1986; Le Brocq et al., 2006). Also, these filters have an abrupt cut-off point where the weighting becomes zero (Le Brocq and others, 2006). Furthermore, effects of perturbations in for example ice thickness have been demonstrated to fade out exponentially for ice flow (Kamb and Echelmeyer, 1986). Because of all these reasons, we apply a local exponential decay filter here (Eq. 6):

$$\nabla.\vec{q}_{[x,y]} = \sum_{i=-dist}^{dist}\sum_{j=-dist}^{dist} W_{[x+i,y+j]}\nabla.\vec{q}_{[x+i,y+j]} \qquad (6)$$






Here, dist is the maximum distance of a cell which is taken into account in the exponential decay filter (set at 2.5
km), i and j are indices, and W represents the weight of a particular cell at position [x+i, y+j] (Le Brocq et al.,

375 2006):



$$W_{[x,y]} = e^{-\frac{1}{sl} * \sqrt{(x'-x)^2 + (y'-y)^2}}$$ (7)


In Eq. 7, x and y are the coordinates of the point being filtered while x' and y' are the coordinates of the weighted
points, sl is the scaling length and is a crucial parameter determining how fast the exponential filter fades. This
parameter directly relates to the length scale over which the longitudinal stresses determine the local ice flow.
Theoretically, this scaling length is in the range of 1-3 times the local ice thickness for valley glaciers and 4-10
times for ice sheets (Kamb and Echelmeyer, 1986; Le Brocq et al., 2006). For our experiments, we vary the scaling
length depending on A times the local ice thickness (Eq. 8). In this way, we incorporate variations within the study
area into the exponential decay filter and we account for non-local flow coupling. The latter was shown to be an
improvement compared to fixed-size filters (Le Brocq et al., 2006). In this way, the ice flux divergence in areas
with a larger ice thickness is considered over a larger distance compared to areas with a smaller ice thickness.


$$sl[x,y] = A * H[x,y] \text{ with } A \in \{0:1:10\}$$ (8)


To determine the optimal procedure to include the effects of (longitudinal) stresses and uncertainty in the input
data, we perform multiple experiments with the exponential decay filter. For this, we examine whether (i) the ice
flux divergence, (ii) the gradients of velocity and ice thickness or (iii) both should be considered over larger spatial
scales for optimal results in the determination of the SMB.

First, the ice flux divergence is considered over larger spatial scales by applying the exponential decay filter to the
ice flux divergence field. Second, to compensate solely for the effects of large gradients, the gradients are
considered over larger spatial scales. We therefore apply the exponential filter to the ice thickness and velocity
gradients and calculate the ice flux divergence using these smoothed gradients. Third, to compensate for the
negative effects of very large scaling lengths for both previous experiments and the biases related to both, we filter
twice. Hence, both filters are applied after each other. In this way, the smoothness of the ice flux divergence field
increases while the top values are not damped as much as applying a filter once with a long scaling length. Filtering
twice is essentially similar to first resampling to a lower resolution and then applying one filter. Earlier research
did not require such filtering to be applied due to a much coarser resolution (Seroussi et al., 2011; Rounce et al.,
2018). For every experiment, the modelled and measured SMB values at the position of the stakes are compared
(see section 2.2). As metrics to quantify the performance of the procedure, the mean absolute error (MAE) and the
root mean square error (RMSE) are used (Eq. 9 and 10). The MAE is defined as the absolute difference between

the modelled and the measured SMB. The RMSE on the other hand takes into account the variance of the errors.

n is the number of stake measurements. The uncertainty of the SMB measurements is estimated to be ± 0.2 m i.e

yr$^{-1}$ and is also taken into account in the analysis. More specifically, for each filter option we carry out 100 versions

in which we perturb the measured SMB. This is done by randomly adding a value between -0.2 and 0.2 distributed

around 0. Then, for the option under consideration, the average MAE and RMSE are calculated from Eqs. 9 and

10, where n is the number of SMB measurements for every year which is 16, see section 2.2. i is an index ranging

from 1 to n (16), $x_i$ and $y_i$ are the Cartesian coordinates of the different stakes under consideration.

.

$$MAE = \frac{1}{n}\left( \left| \sum_{i=1}^{n} b_{s,modelled}(x_i, y_i) - b_{s,measured}(i) \right| \right) \tag{9}$$

$$RMSE = \sqrt{\frac{\sum_{i=1}^{n} (b_{s,modelled}(x_i, y_i) - b_{s,measured}(i))^2}{n}} \tag{10}$$

## 4    Results

### 4.1  Surface elevation changes and filtering

The accuracy of the elevation product is assessed by comparing the photogrammetrically created DSMs with

GVPs, randomly divided over the study area. Mean absolute errors (MAE) between measured and modelled

elevation are in the order of a few cm (Table 2), which is similar to values found in other studies (Whitehead et

al., 2013; Immerzeel et al., 2014; Wigmore and Mark, 2017; Zhang et al., 2019). As such, the accuracy of the

created DSMs is high. Furthermore, the mean error (ME) is close to zero which indicates that the created DSMs

are not biased. For 2017, the slightly larger MAE is probably caused by a smaller number of GCPs combined with

a reduced visual content because of fresh snow on the glacier surface. Further, the small MAE and RMSE of the

DSM in 2020 highlight the advantage of using a RTK equipped with an RTK GPS (P4RTK) for which a smaller

amount of GCPs is needed to reach similar or better accuracies compared to a classic setup (UAV without RTK

correction).

*Table 2. Mean absolute error (MAE) and root mean square error (RMSE) of the elevation differences between the DSMs and*
*the GVPs in different years. (Units are in m).*

| Year | MAE | ME | RMSE | GCP density (km$^{-2}$) | UAV used |
|------|-----|-----|------|-------------------------|----------|
| **2017** | 0.09 | -0.07 | 0.16 | 11 | P4PRO |
| **2018** | 0.06 | -0.02 | 0.22 | 24 | P4PRO |
| **2019** | 0.08 | 0.03 | 0.16 | 18 | P4PRO |
| **2020** | 0.07 | 0.03 | 0.10 | 11 | P4RTK |





The different maps for the three balance years, display a very detailed pattern of surface elevation changes (Figure 5). For example, alternating positive and negative surface elevation changes are noticeable and can be explained by the advection of local glacier surface topography due to glacier movement (see also Figure 4). In addition, significant less negative $\partial h/\partial t$ values at the left side of Vadret da Morteratsch, especially where the glacier protrudes towards the north, are likely the result of a debris cover insulating the ice below and decreasing the melting rate (Reznichenko et al., 2010; Rounce et al., 2018). Given that the ice fluxes can be assumed to remain relatively constant over the three years in consideration, the elevation changes indicate that 2017-2018 was the year with the most negative SMB (which is confirmed by field measurements, see e.g. Fig. 12). Further, large positive surface elevation changes can be distinguished for 2019-2020 at the highest part of Vadret Pers. This is probably the result of avalanches originating from the steep accumulation area and persistent snow cover in these areas. This is striking as surface elevation changes further down on the glacier are generally more negative compared to 2018-2019: i.e. the SMB gradient on Vadret Pers is steeper in 2019-2020 compared to 2018-2019.

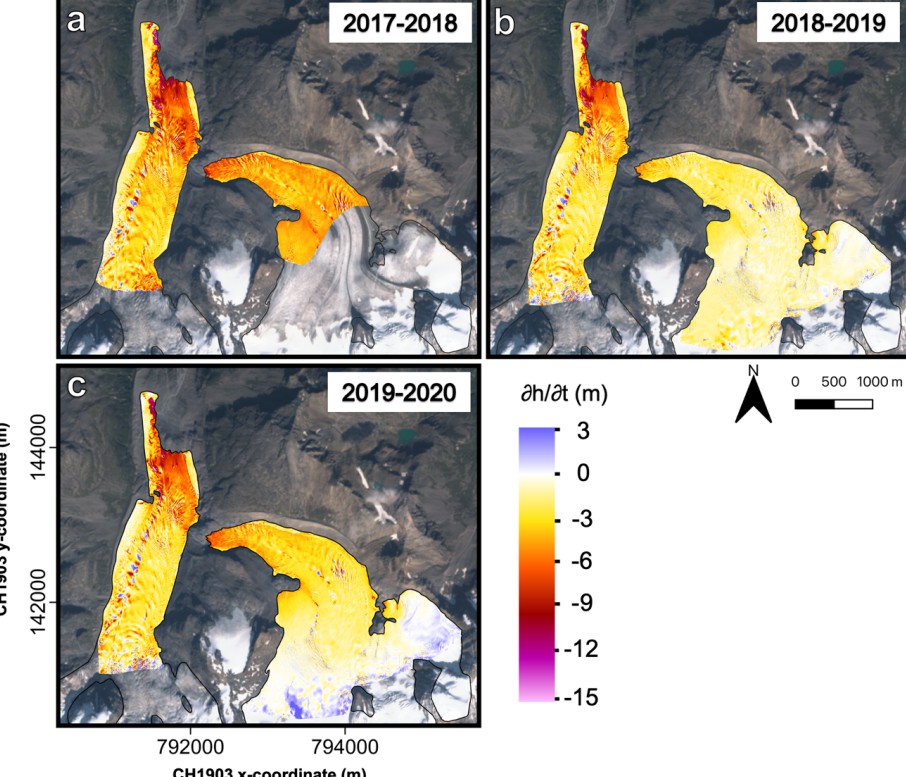

*Figure 5. Surface elevation changes for (a) 2017-2018, (b) 2018-2019 and (c) 2019-2020. The spatial resolution is equal to 8 cm. The outline of the glacier corresponds to the latest year of observation for every balance year in consideration. The background image is a Sentinel-2 true colour composite satellite image from 13 September 2020.*







Alternating positive and negative surface elevation changes are occurring, caused by the advection of surface
topography (Figure 4 and Figure 5). We remove these non-SMB features by applying a block moving average
filter over a distance that is determined by the local surface velocity (see section 3.2). When the smoothing
parameter (k-value) is increased, the standard deviation (SD) within elevation bands of 100 m decreases
significantly for all elevation intervals above 2300 m (Figure 6). This means that the variation is considerably
reduced within these bands. The largest decrease is for k-values between 0 and 1 but only from a k-value of 3
onwards, an almost complete levelling out of the SD can be noticed. In addition, it is clear that the SD remains
generally above 0.5 m. This is due to variations within the elevation bands that are preserved, for example caused
by elevation differences.


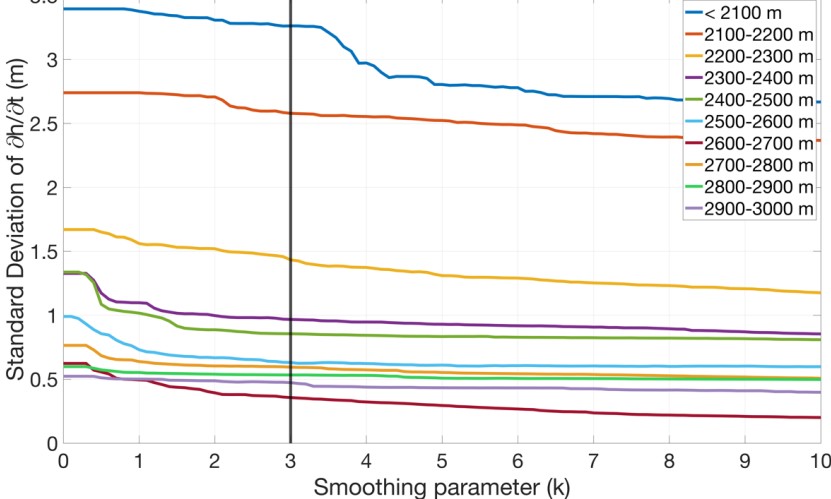



**Figure 6.** *The surface elevation changes are smoothed using a block moving average filter over a distance depending on the*
*local surface velocity and a smoothing parameter (k) between 0 and 10. The different lines represent the standard deviation*
*(m) of the surface elevation changes within elevation bands of 100 metres. The vertical black line indicates the chosen*
*parameter of k = 3.*


Further, for the elevation bands below 2300 metres, the SD remains higher and decreases more slowly or even not
at all. These elevation bands concern the lowest, slow flowing glacier areas of Vadret da Morteratsch. Here,
variations in surface elevation changes are not related to the advection of surface topography and are correctly
kept (not smoothed). The variations in this area are for example caused by differential melting due to debris. If a
constant box filter would have been used, this would have resulted in the surface elevation changes in these areas





to be smoothed to the same extent. This would imply that actual SMB caused patterns would no longer be visible, which highlights the advantage of using a filter with a window size depending on the local surface velocity.

For k=1, there remain a lot of patches with surface elevation changes arising from the glacier movement (Figure 7). This is significantly less when k equals 3. If k were to be enlarged further, for example to k=10, the pattern of $\partial h/\partial t$ fades completely. The positive surface elevation changes in the upper area of Vadret Pers are for example completely smeared out. It is therefore essential to minimise the window size of the block filter so that variations owing to glacier movement are eliminated without losing too much detail. For all these reasons, we choose k to be equal to 3 for further calculations.



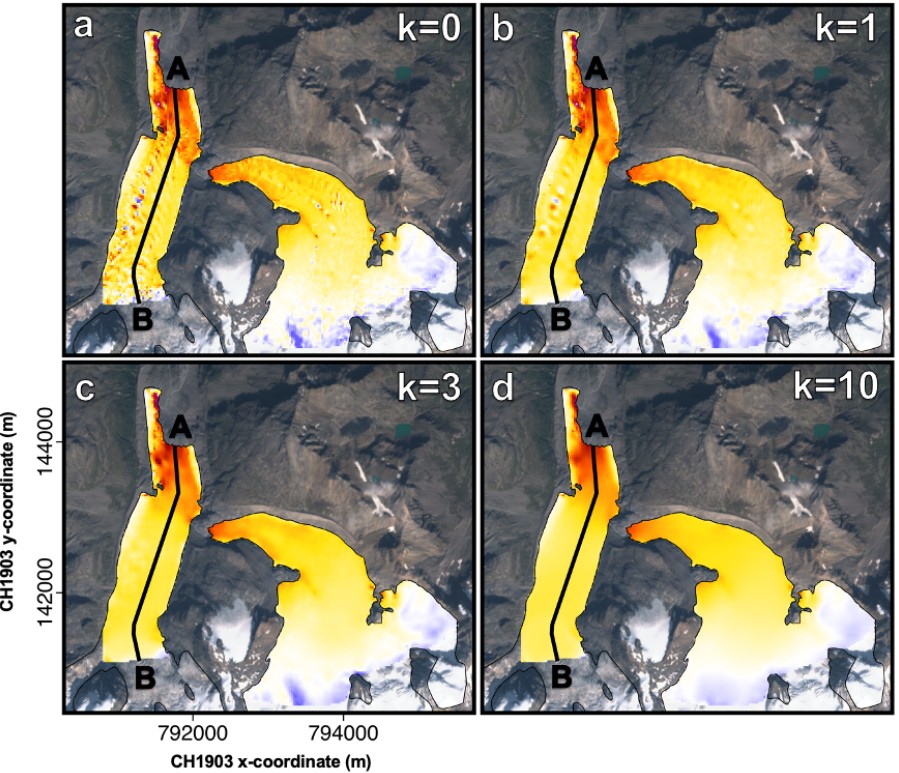


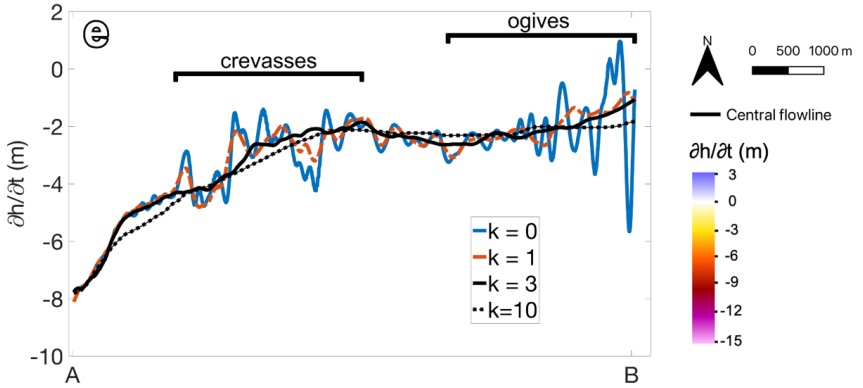


**Figure 7**. *Surface elevation changes between 2019 and 2020, smoothed using a moving average filter and a smoothing*
*parameter (k) of 0 (panel a), 1 (panel b), 3 (panel c) and 10 (panel d). The black line represents an approximation of the central*
*flowline of Vadret da Morteratsch. The background image is a Sentinel-2 true colour composite satellite image from 13*
*September 2020. The graph (panel e) shows the surface elevation changes along the central flowline indicated in panels a, b,*
*c, and d and some distinct topographical features which need to be filtered out.*

## 4.2 Glacier surface velocity

The velocity map derived from feature tracking and filtered for errors (see section 3.3), contains several voids (Figure 8a). This is caused by a lack of correlation between the two years, which is typically related to deformations of the surface and the presence of snow cover or debris. The latter hampers an accurate tracking between two consecutive years. To fill these gaps, spline interpolation is used (Figure 8b).

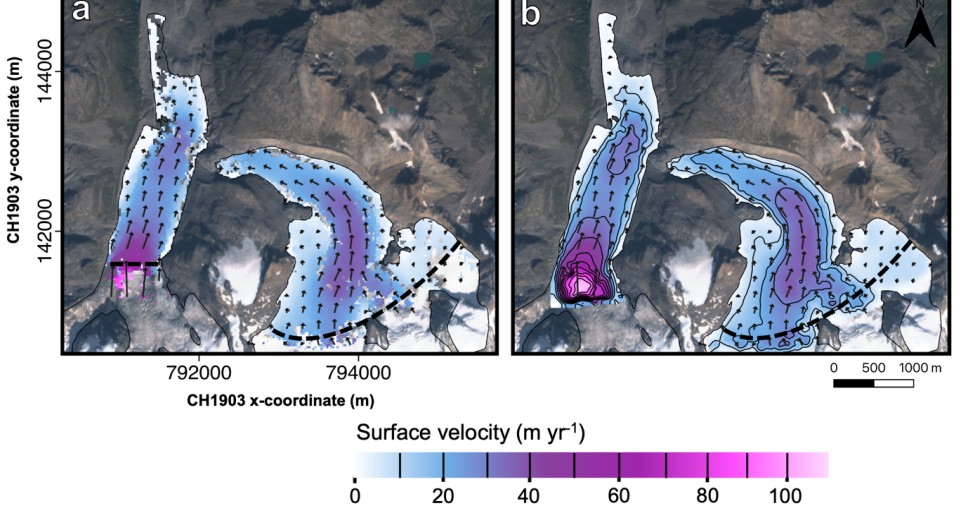

*Figure 8. Horizontal surface velocities derived from feature tracking between 2019 and 2020. (a) filtered surface velocity, (b) void-filled filtered surface velocity. The black arrows indicate the flow direction. The dashed black line indicates the limit of the glacier area where the velocity is modelled without many gaps (area used for further analyses, e.g. when calculating ice flux divergences). The background image is a Sentinel-2 true colour composite satellite image from 13 September 2020. The contour lines are added for every 10 m and they are indicated in the colorbar.*

Comparison with the velocities obtained by field measurements using ablation stakes (based on high-precision RTK GPS system, estimated accuracy of approx. 0.1 m yr$^{-1}$) reveals a good agreement. We find the ME and RMSE to be 0.01 / 1.7 m yr$^{-1}$ for 2017-2018, 0.99 / 1.9 m yr$^{-1}$ for 2018-2019 and -0.37 / 1.8 m yr$^{-1}$ for 2019-2020 (Figure 9).

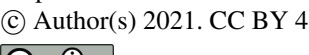



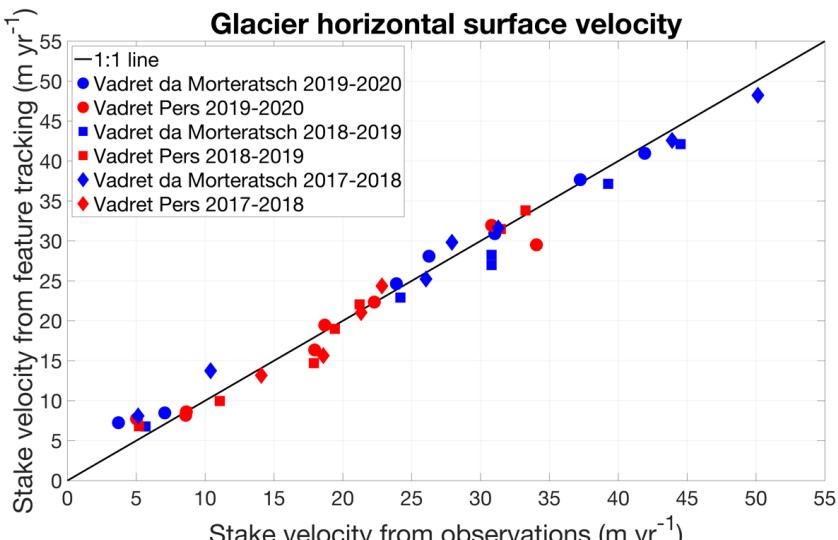

Figure 9. *Comparison between surface velocities at stakes from field observations (high precision GPS measurement at stakes) and from feature tracking.*

The surface velocities of 2017-2018 are higher compared to 2018-2019. This might be caused by the slowdown of the glacier's ablation tongue, commonly observed in mountain valley glaciers as a result of glacier thinning (Dehecq et al., 2019). However, because the surface velocities in 2019-2020 also appear to be slightly larger compared to 2018-2019, it might be due to a larger amount of basal sliding. The latter can be caused by an increased supply of meltwater lubricating the glacier's base which can be linked to a more negative $\partial h/\partial t$ in 2017-2018 and 2019-2020 compared to 2018-2019 (see also Figure 5).

### 4.3 Ice flux divergence

After considering various spatial scales and filter procedures to determine the ice flux divergence, we find optimal results (lowest MAE, RMSE and minimal scaling length, see section 5.2) when both filters are applied after each other for a scaling length of 4xH to smooth the ice thickness and velocity gradients and a scaling length of 1xH to smooth the ensuing ice flux divergence (see section 4.3). These values are close to the theoretical values of the fundamental longitudinal scaling length for valley glaciers, mentioned in Kamb and Echelmeyer (1986). It shows that the data needs to be considered over large spatial scales which implies that the ice flux divergence field must be sufficiently smooth to give accurate results.

The computed ice flux divergence is clearly characterized by negative values over both glacier tongues (Figure 10). This represents horizontal compression and associated vertical extension which is generally expected for



ablation areas (e.g. Kääb and Funk, 1999). This horizontal compression translates into an ice supply towards the
surface (emergence velocity), which counters the effect of the negative mass balance on the local ice thickness
change (and neutralizes it in case the glacier is in equilibrium with the local climatic conditions). The ice flux
divergence reaches a minimum between 500 and 1000 m from the terminus of the Vadret da Morteratsch with
values close to 4 m i.e. yr$^{-1}$ (Figure 10). In this area, both the ice thickness (Figure 3) and the surface velocity
(Figure 8) decrease significantly which leads to large negative gradients and hence upward motion (negative ice
flux divergence). It is therefore not surprising that there is a complex pattern of transverse crevasses in this area
(see Figure 2). The maximum ice flux divergence at the eastern side of Vadret da Morteratsch near the former
confluence area with Vadret Pers is suspicious. This is likely caused by the very large ice thickness gradient as a
result of underestimated ice thickness in the THIZ dataset. In this area, the THIZ dataset revealed zero ice thickness
as a result of an overestimation of the bedrock elevation (Figure 3).


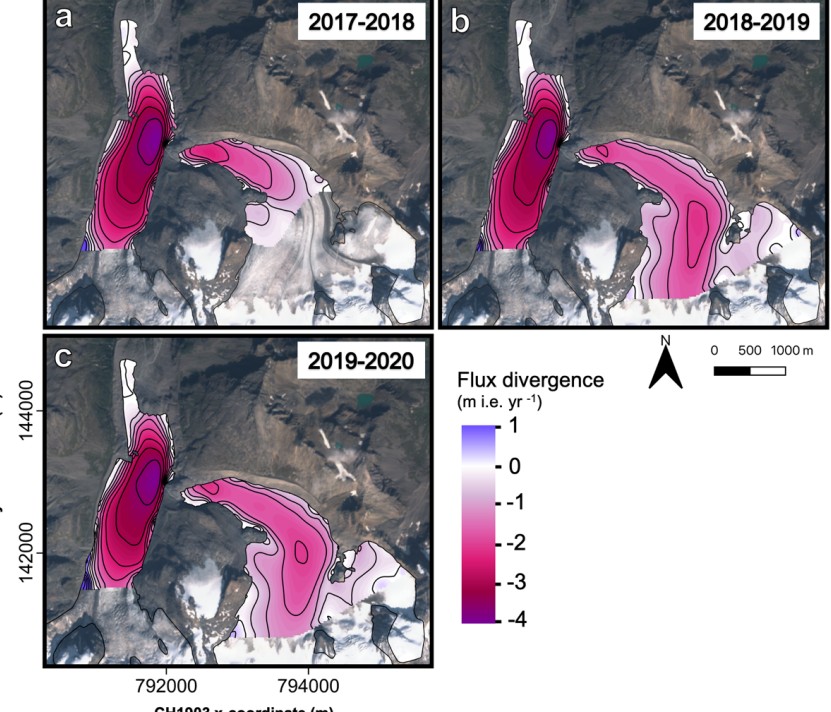



**Figure 10.** *Maps of the modelled ice flux divergence for the different years under consideration. Contours are added for every*
*0.5 metre per year. The background image is a Sentinel-2 true colour composite satellite image from 13 September 2020.*






### 4.4 SMB from UAV


The final step of the method is to add the surface elevation changes (see Figure 5 and Figure 6) and the computed ice flux divergence (Figure 10) to obtain the SMB (see Eq. 1). The final product, the modelled SMB for the three balance years, is shown in Figure 11. The three years in consideration clearly show a similar pattern. Areas with a less negative SMB are located at the same locations (e.g. western margin of Vadret da Morteratsch). Areas with the most negative SMB are also at the same location (e.g. glacier fronts). The ice flux divergence is prominently not able to compensate for the very negative SMB so that there is net mass loss everywhere and the altitude changes are less than zero. The most negative SMB (up to -13 m i.e. yr$^{-1}$) is found at the front of Vadret da Morteratsch in 2017-2018 which is in accordance with the SMB measurements. This concerns the lowest area which is also quasi-stagnant with a very limited supply of ice from upstream (see Figure 10).

592

593

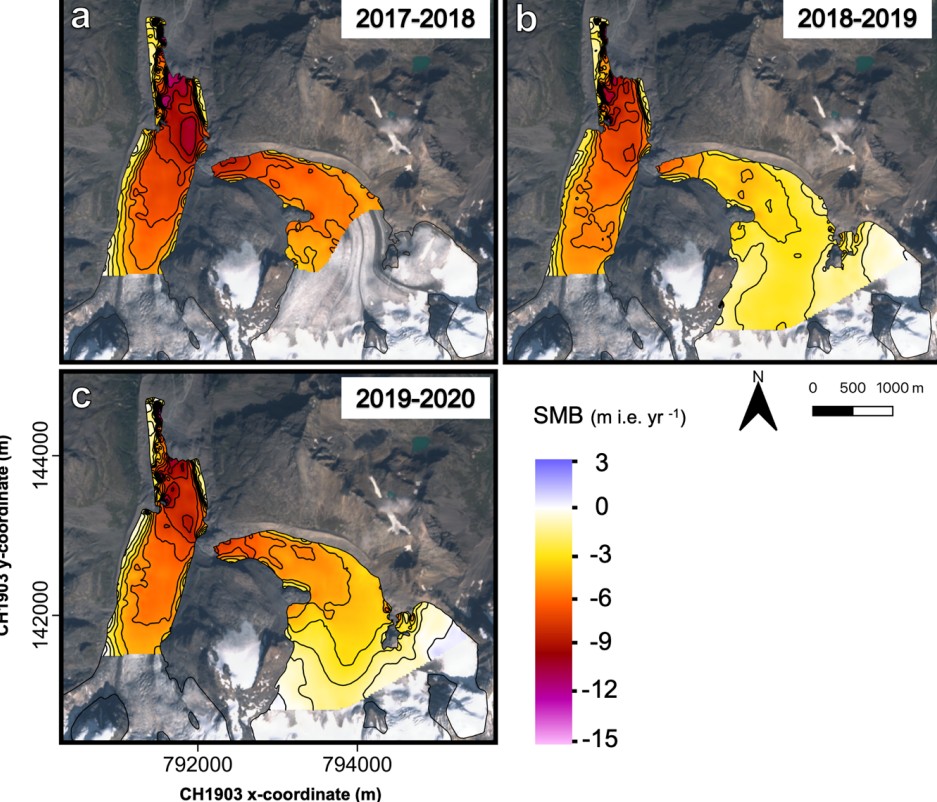

**Figure 11.** *Maps of the modelled SMB fields. Contours are added for every metre per year. The background image is a Sentinel-2 true colour composite satellite image from 13 September 2020.*



The associated point-by-point comparison between modelled and measured SMB confirms the good match with deviations generally smaller than 0.5 m i.e. $yr^{-1}$ (Figure 12). The largest differences are found at the front of Vadret Pers (Figure 12). Near the glacier front, the ice flux divergence is likely underestimated as a result of overestimated ice thickness in this area of the THIZ dataset. The latter decreases the ice thickness gradient in this area. Stake M26 (see Figure 2) is excluded from the analysis because it was located in the middle of a field of transverse crevasses during the survey in 2018. This location might be associated with increased ablation because the stake was located on the top of a serac.

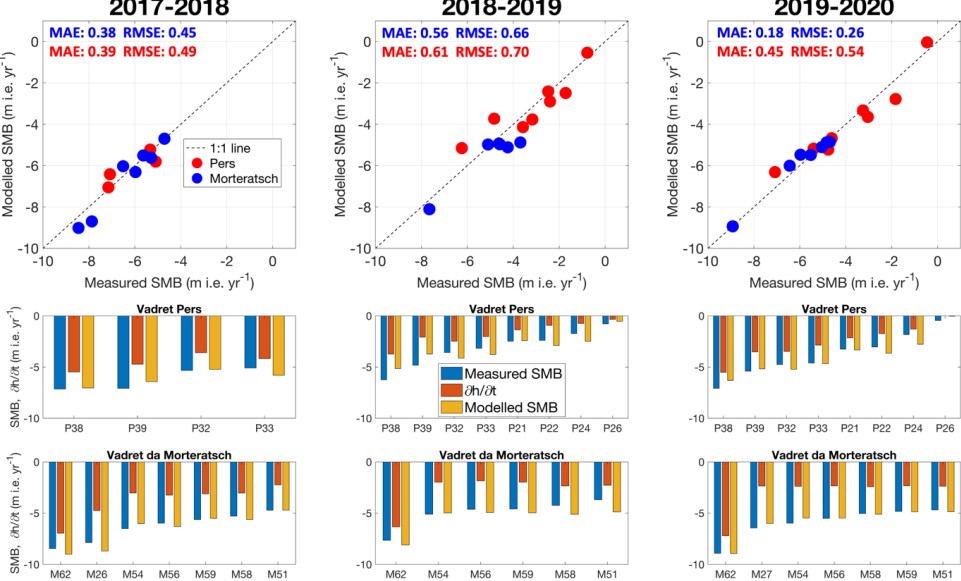

***Figure 12.*** *Point-by-point comparison of modelled and measured SMB. In the lower panels, the difference between the modelled SMB (yellow bar) and the local elevation change (red bar) results from the modelled ice flux divergence. The labels refer to the stake names as used in the fieldwork programme. The MAE and RMSE of the modelled SMB are added in the upper panels (Values are in m i.e $yr^{-1}$).*

The MAE and RMSE for the optimal SMB fields are well below 1 m i.e $yr^{-1}$ with a minimum of 0.18 m i.e $yr^{-1}$ for Vadret da Morteratsch in 2019-2020 (Figure 12). The MAE and RMSE are somewhat larger for Vadret Pers with a maximum of 0.61 m i.e $yr^{-1}$. in 2018-2019 which is mainly caused by the larger deviation for the lowest two stakes (Figure 12).





### 4.5 Lateral variations in the SMB pattern

Using the continuity method based on close-range (UAV) remote sensing data clearly reveals a much more detailed picture of SMB than is possible from a stake network on a limited number of locations. This is evident in the lateral heterogeneity of SMB, which is often overlooked when considering elevation as the prime variable to plan the stake locations. In Figure 13 the difference is shown between the UAV-derived SMB field and a SMB field only determined by elevation as derived from a linear fit of the stake measurements with altitude. In general, the largest differences occur close to the margin of the glaciers, and around their front. The differences at the glacier margin of Vadret da Morteratsch are mainly related to a thick debris cover, which when sufficient in thickness, has an insulating effect that reduces the glacier melt (e.g. Rounce et al., 2018; Verhaegen et al., 2020). For example, for the heavily-debris covered area where Vadret da Morteratsch protrudes towards the north, the SMB is -1.5 m i.e. $yr^{-1}$ below the debris and up to -12 m i.e. $yr^{-1}$ for the clean ice next to the debris. The melt ratio is therefore equal to 0.125, which means that there is 87.5% less melt under the debris at this location. This corresponds to a debris thickness of more than 50 cm when a typical average value of the characteristic debris thickness is used (Anderson and Anderson, 2016; Rounce et al., 2018). Such thicknesses are very likely in this area where boulders from the lateral moraines supply supraglacial debris.

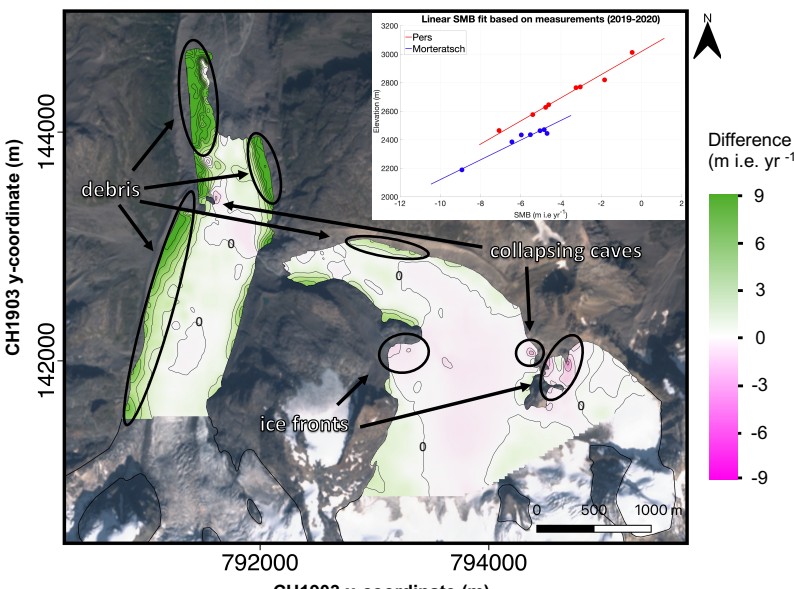

**Figure 13.** *The inset shows a linear SMB fit based on stake measurements. The main figure shows the difference between the UAV-derived SMB and the linearly derived SMB. The background image is a Sentinel-2 true colour composite satellite image from 13 September 2020. Contour lines are added for every metre. Labels are added for the contour of value 0 m i.e. $yr^{-1}$.*




Another major feature is the presence of patches with a clearly more negative SMB when relying on the UAV-
derived SMB field. These differences relate to stationary areas at local ice fronts or collapsing ice caves. While
the first are real SMB variations (caused by more irradiation and/or less snow), the latter ones are not caused by
the SMB. These collapsing ice caves clearly appear to occur near stagnant ice and at the bottom of the glacier
where melt water forms a cave below the ice (Figure 13b). These caves collapse when the overlying ice becomes
too thin due to melting at the surface. For the other areas, the difference between the SMB fields is generally
smaller, between -0.5 and 0.5 m i.e yr$^{-1}$. This is in the same order as the MAE and the RMSE and is slightly larger
than the expected accuracy of the SMB measurements.


**5    Sensitivity analysis of the selected data, parameters and filters**

The surface elevation changes and the ice flux divergence contribute separately to the surface mass balance,
implying that errors and uncertainties in both terms do not influence each other but directly affect the determined
SMB (Eq. 4). It is therefore crucial to determine to which extent the uncertainty of both affects the results of the
applied method. In addition, we evaluate the results of the different filter procedures of the ice flux divergence.

**5.1  Dependence on the data and F-value used**

To study the sensitivity of the applied method to the used data and F-value, we create 100 perturbed fields of the
surface elevation changes, the surface velocity, the F-value and the ice thickness. After that, we do the calculations,
using the perturbed data for one dataset and the original data for the other ones. Next, the MAE of modelled versus
measured SMB is calculated 100 times and the standard deviation of this MAE is determined. The uncertainty in
the surface elevation changes is assumed to be correlated in space, i.e. depending on a pattern across the glacier
(e.g. Fisher et al., 2015). Therefore, we perturb the $\partial h/\partial t$ fields using randomly spread patches with diameters
between 50 m (local errors at grid level) and 500 m with perturbations of maximum 0.5 m (plausible maximum
value for UAV data). Regarding surface velocity, we use error estimates of 10% of the observed velocity, similar
to the expected accuracy (see section 4.2). Besides velocity itself, the assumption of a constant F-value is rather
unrealistic (Zekollari et al., 2013). Ice flow over basal irregularities causes variations in F. Previous research
showed for example F to be larger over basal highs and smaller over basal lows and glacier areas with more basal
sliding might as well be characterized by a larger F-value (Reeh et al., 2003). Therefore, we apply perturbations
using an F-value between 0.85 and 0.95. Concerning ice thickness, we apply perturbations of 30% of the local ice
thickness which is mentioned by Zekollari et al. (2013) as an estimate of the accuracy.





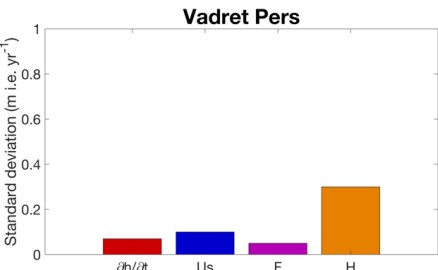
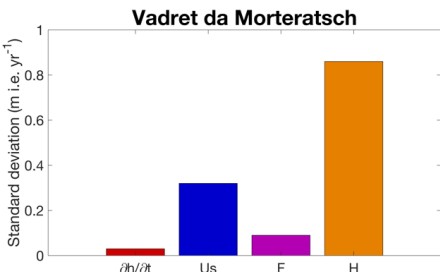

**Figure 14**. *Standard deviation of the MAE for the perturbed versions of the surface elevation changes, the surface velocity ($u_s$), the F-value and the ice thickness (H).*

The effect of perturbation of $\partial h / \partial t$ on the determined SMB is small with a standard deviation of less than 0.1 m i.e. yr$^{-1}$ for both glaciers (Figure 14). As a result, the applied method is not very sensitive to (small) perturbations in $\partial h / \partial t$. Changes in the F-value and surface velocity also appear to have minor sensitivity, especially concerning Vadret Pers. Regarding Vadret da Morteratsch, the surface velocity has a SD of 0.3 m i.e. yr$^{-1}$, which is not negligible (Figure 14). The ice thickness distribution, however, is for both glaciers undoubtedly most crucial for the determination of the SMB, and even more so for Vadret da Morteratsch.

## 5.2 Evaluation of procedures to filter the ice flux divergence

As the ice flow at a given location is determined by the surrounding glacier geometry (i.e. not only by its local geometry) and to avoid non-physical oscillations in the flux divergence field (as a result of fine grid spacing), it appears to be necessary to consider various spatial scales over which the ice flux divergence needed to be determined and different filtering procedures (see section 3.4). The results of each possible combination are shown in Figure 15.

By only applying the exponential decay filter to the ice flux divergence field (moving down in the matrix), we find minimum MAE and RMSE values when the scaling length is equal to seven times H (MAE for Morteratsch-Pers is 0.55 m i.e. yr$^{-1}$). Such a large scaling length indicates that the ice flux divergence field must be smoothed significantly and therefore becomes entirely smeared with limited variation in the ablation area. One of the reasons for this high value are large ice thickness and velocity gradients resulting from solving ice flow processes on a high-resolution numerical grid. To compensate for the effects of large gradients, the second option is to consider the gradients over larger spatial scales. We do this by applying the exponential filter to the ice thickness and velocity gradients and calculate the ice flux divergence using these smoothed gradients (moving to the right in the matrix). Here, the MAE remains in all experiments above 0.6 m i.e. yr$^{-1}$. In addition, this value is only reached as soon as the velocity gradient is considered over ten times H and the ice thickness gradients over five times H. The solution to compensate for the negative effects of a very large scaling length for both previous filters and the biases related to both is to filter twice, as was shown in section 4.3. As soon as the ice thickness and velocity gradients





and the ice flux divergence are smoothed, the MAE becomes significantly smaller (Figure 15). The MAE is now
much below 1 m i.e. yr$^{-1}$.


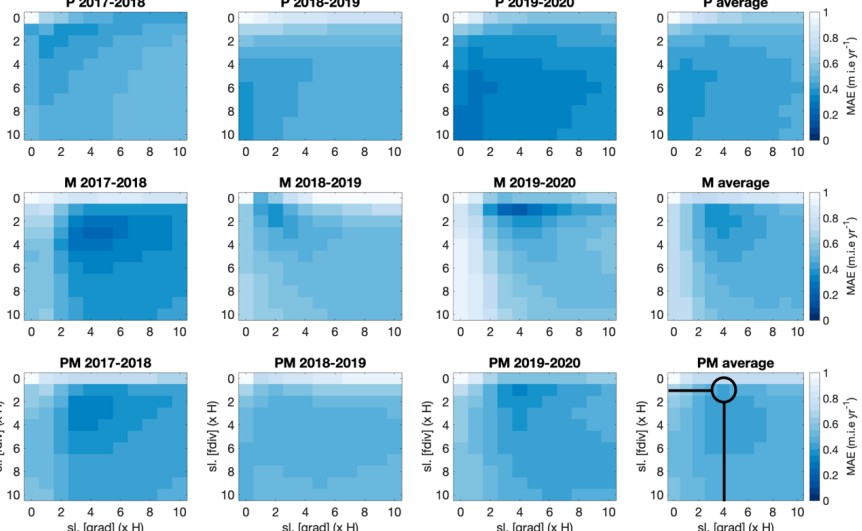



***Figure 15.*** *Mean absolute error between measured and modelled SMB for various scaling lengths. The scaling length depends*
*on the local ice thickness (x H). Smoothing the ice thickness and velocity gradients (grad) or smoothing the ice flux divergence*
*(fdiv) is represented by the horizontal axis and vertical axis, respectively. M represents Vadret da Morteratsch, P represents*
*Vadret Pers and PM represents the entire glacier complex. The plots entitled with average concern the average over all balance*
*years. The selected combination with which the ice flux divergence is calculated as shown in Figure 10 is encircled with black.*


Another observation standing out is that concerning Vadret Pers, the MAE mostly decreases when the ice flux
divergence is smoothed while for Vadret da Morteratsch, smoothing the gradients has a larger impact (Figure 15).
Our explanation for these peculiar differences is that the gradients (both in terms of ice thickness and surface
velocity) for Vadret Pers are already significantly smaller compared to those of Vadret da Morteratsch. The latter
glacier has both a higher velocity and a larger ice thickness. Consequently, smoothing of the gradients for Vadret
da Morteratsch is more decisive, whereas for Vadret Pers it is the other way around.


**6   Conclusions**

In this study, a method was presented to estimate the surface mass balance pattern from UAV observations, a
known ice thickness field, and the principles of mass continuity. Annual surface elevation changes and surface
velocities were quantified from UAV and were shown to have centimetre accuracy. The method was applied to
the entire ablation zone of Vadret da Morteratsch and Vadret Pers (Switzerland) for three individual balance years





between 2017 and 2020. For the well-studied Morteratsch-Pers glacier complex, we were able to closely reproduce SMB surveyed at about 16 stake locations. A major advantage of using close-range (UAV) remote sensing data is that a significantly more detailed SMB pattern can be obtained over the entire ablation area at a high spatial resolution, which is not feasible with a (small) number of stakes. The need for inter- and extrapolations from a limited number of stakes can introduce several significant errors when the heterogeneity of the glacier surface cannot be captured well from the installed stake positions.

The analysis did not demonstrate a simple consensus on how to consider the ice flux divergence for optimal results on both glaciers. It proved to be necessary to consider large spatial scales concerning the ice flux divergence to apply the continuity-equation method and closely reproduce the SMB. By using an exponential decay filter to the ice thickness and velocity gradients and the ensuing ice flux divergence over several times the local ice thickness (at least 2x), the mean absolute error between modelled and measured SMB decreased to 0.3-0.7 m i.e. $yr^{-1}$ on average for the three balance years. Considering the uncertainty of the data and the stake measurements, this is quite accurate.

The applicability of the applied method to other, less well studied glaciers, depends on how closely the elevation changes, the surface velocities and the ice thickness can be estimated for these ice bodies. In recent years, a new series of methods have arisen from which surface elevation changes and surface velocities can be derived over glaciers (Brun et al., 2017; Paul et al., 2017; Millan et al., 2019; Braun et al., 2019; Nagy et al., 2019; Dussaillant et al., 2019; Sommer et al., 2019). This seems encouraging to investigate the applicability using high-resolution satellite data, preferably lower than 10 m for accurate velocity determination. However, our analysis also showed that especially the ice thickness should be well known. The latter seems to hamper the application of the method to many glaciers because large-scale ice thickness estimates are often not accurate enough. As shown in a recent study on four Central Asian glaciers such large-scale thickness estimates may capture the general pattern of the ice thickness distribution and total volume well, but they exhibit significant deviations at the local scale (Van Tricht et al., 2020).

**Code and data availability**

Model code written in Matlab and DSMs will be provided on Github.

**Author contribution**

LVT developed the method, performed the experiments and wrote the manuscript. PH provided guidance in implementing the research and interpreting the results and assisted during the entire process. JVB and AV contributed to the fieldwork, collaborated in developing the method and improved the manuscript throughout the entire process. KVO assisted during the fieldwork and introduced LVT in using UAVs for research purposes. HZ



participated in the fieldwork for many years and contributed throughout the entire process to developing the
method and optimising and refining the research.


**Competing interests.**

The authors declare that they have no conflict of interest.


**Acknowledgements**

The authors would like to thank Chloe Marie Paice, Felix Vanderleenen, He Zhang, Robbe Neyns, Steven De
Hertog, Veronica Tollenaar and Yoni Verhaegen, who assisted during the fieldwork to perform stake
measurements and distribute and collect GCPs.


**Financial Support**

Lander Van Tricht holds a PhD fellowship of the Research Foundation-Flanders (FWO-Vlaanderen) and is
affiliated with the Vrije Universiteit Brussel (VUB). Harry Zekollari contributed to the fieldwork as a PhD fellow
of the Research Foundation-Flanders (FWO-Vlaanderen) and at a later stage as a Marie Skłodowska-Curie fellow
at the TU Delft (grant 799904). K Van Oost is an FNRS Research Director.





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
