# Peer review of "Estimating surface mass balance patterns from UAV measurements"

_The Cryosphere, 2021_

## Author Comment (AC1)

In this document, we respond to the comments of reviewer 1 one by one. Whenever some entirely new text has been added to the manuscript, it has been added in italics and in red below.

In the proposed revised manuscript, which is added at the bottom of this document, the textual changes have been added in red.

**Reviewer 1 – Evan Miles**

The study by Van Tricht et al presents a set of experiments to find the best image processing approach to resolve glacier surface mass balance from high-quality multi-year UAV measurements. The study is generally well implemented and seems to produce an approach to derive reasonable distributed SMB estimates based on the correspondence with stake measurements, suggesting filtering of thickness and velocity gradients, as well as direct flux divergence values, over a multiple-ice-thickness length scale. The study also highlights the importance of high-accuracy ice thickness data.

This is a welcome contribution to the field, as implementations of the continuity approach have come increasingly into vogue over the past few years, and UAV surveys have made high-precision topographic data a routine aspect of glacier monitoring. This study particularly offers the potential of deriving spatially-distributed flux divergence values, rather than based on glacier segments or for a single point. It is especially nice that authors have made use of a great set of field data to evaluate their model well; this is an important and challenging step. The manuscript is nicely written and the work is presented well. I do have a number of comments for the authors, including a few more substantial methodological concerns which the authors should consider, particularly with regards to the choice of filters and assumptions/effects thereof.

We would like to thank the reviewer for taking the time to provide a useful, comprehensive, and detailed review of our paper. We have addressed all comments below and updated the manuscript accordingly where needed. We believe this has strongly improved the quality of the research.

**General comments:**

**[RC1.GC1]** It is not apparent that the authors have evaluated whether the filtering has significantly impacted the total net volume change (or net flux divergence) – is mass conserved after all the filtering? Essentially, neither the variable box average filter or the exponential decay filter are conservative – they change the local and non-local mean values. I think it is likely that this does not result in a significant difference in the total volume loss or total flux divergence, but the authors should evaluate whether the difference falls within the uncertainty of the underlying data, or whether the filtering has internally broken mass conservation.

The reviewer raises an important subject here. We have addressed this in the revised manuscript. To provide for conservation of mass concerning the ice flux divergence, we ensured that the net

flux divergence before and after the filtering is the same by multiplying the filtered values by the ratio of the net flux divergence before and after filtering. This information is added in the text in lines 409-410:

*"To provide for conservation of mass, the filtered result in each grid cell is multiplied by the ratio of the net flux divergence (sum of all ice flux divergences) before and after filtering."*

It is worth noting that the differences are small, so the results are minimally affected. The ratio is 0.93 for 2-17-2018, 0.91 for 2018-2019 and 0.92 for 2019 -2020.

**[RC1.GC2]** I found it strange for all values to be in ice equivalent! This avoids the difficult topic of density (less difficult in the ablation area) but I think it would be better to make an assumption, present your results in the standard unit, and then briefly mention this problem in the discussion (future work/opportunities/needs).

We understand this comment of the reviewer. However, the direct unit inferred from our approach is the surface density. In the ablation area, where the surface consists entirely out of ice, the latter is ice density. Further, because in the continuity equation, two out of three terms are in ice density, we kept the units in ice density. A conversion to water equivalent would require a single ice density assumption for the area under consideration. We added a section in the discussion in which we elaborate on the density considerations in lines 768-772:

*"To convert the results presented here to the conventional unit of mass balance, assumptions must be made for density. In the ablation area, to which our study area was limited, the SMB could have been converted directly to metres of water equivalent using an average ice density such as 900 kg/m3. However, when the presented method would be applied to for example the accumulation area, specific attention is required for each quantity of the continuity equation (Miles et al., 2021)."*

**[RC1.GC3]** One of the filtering steps (for the dH/dt data) aims to correct apparent surface changes due to advection of surface features. I would recommend that the authors examine the flow corrections of e.g. Brun et al (2018) which are a more appropriate approach to resolve this problem, but requires switching from an Eulerian to Lagrangian frame, which is anyhow more appropriate for comparing to a stake.

We are aware that with our approach we may have filtered out also relevant measured changes. Because the method of flow correcting the DSMs, as for example used by Brun et al. (2018), is physically more justifiable, we agree with the different reviewers to apply this method. We added the explanation in lines 287-291:

*"The different DSMs are treated considering the motion of the ice, following the method applied in e.g. Brun et al. (2018). This method consists of projecting each point on the glacier back to its original position in the previous year, based on the local velocity in x and y direction and taking into account*

*the vertical displacement produced by flow along the longitudinal slope. The latter is determined by taking the average slope over all the different years for which a DSM was made, smoothed with a Gaussian filter."*

The results have changed little, perhaps because the stakes are in stable areas of the glacier, so the flow corrections and our filter method do not differ much. However, the pattern of the SMB has become less homogeneous, due to artefacts in the flow correction method. This is clarified in the text in lines 596-600:

*"The presence of certain irregular patches with a smaller or larger SMB is caused by artefacts in the flow correction method (see section 3.2 and Figure 6d). This is because the flow correction method cannot trace all displacements exactly (rocks fall over, crevasses deform, supraglacial melt streams are positioned on different locations etc.), and because the velocity field is not fully accurate, and the calculation of the longitudinal slope also affects the result."*

**[RC1.GC4]** For both velocity and thinning datasets, the authors have discarded data outside the glacier boundary, but they should instead use these data to empirically assess the quality of their results. Similarly, the combination of the ice thickness datasets is a bit awkward; it would make more sense to me to integrate all available bed elevation observations to produce an optimal ice thickness dataset. However, this may not be feasible for multiple reasons, and would require reprocessing nearly all datasets, so I can understand that the authors may wish to avoid that course.

Using the areas outside the glacier outline is definitely a useful way to measure the absolute error, in addition to the use of validation points which can specifically be used to measure the relative error and self-consistency of the DSMs. In the section where we describe the error based on the validation points (section 4.1), we added a text and figure in which we elaborate on the off-glacier variations in surface elevation changes (lines 468-477):

*"Then, the accuracy of the surface elevation changes is assessed by comparing the elevation differences in rectangular areas outside the glacierized areas for two regions (see Figure 6), where the topography is assumed to be mostly stable (Yang et al., 2020). The histograms for the three different balance years in consideration show that most of the differences range between -0.2 and 0.2 m, which indicates that there is no large absolute offset between the DSMs of the different years (Figure). For region 2, we find a larger standard deviation (SD) for the 2017-2018 difference which appears to be caused by a small horizontal shift which produces larger elevation differences (both negative and positive) in this steeper area. For the gentler sloping glacier surface, the effects are negligible. The larger SD for the 2019-2020 difference in region 1 is caused by an area where a part of the bare surface was eroded by a melt river, producing a tail of negative elevation differences. In general, the above analysis reveals that the vertical uncertainties resulting from the DSM differencing are limited."*

[Figure]

*Figure 5. Histograms of elevation differences for the different balance years considered. The mean error (ME), median error (MED), and standard deviation (SD) are added for each region.*

as well as concerning the surface velocity in line 310:

*"after verifying average zero displacement for stable areas outside the glacier"*

Concerning the ice thickness datasets, indeed, both datasets differ substantially, although the patterns are similar. Our measurements in 2020 (which lie between the two datasets) showed that neither one of the two could be dismissed. Therefore, we opted to take the average ice thickness as our main dataset for this study. Furthermore, extensive tests with both datasets encouraged us again to use the average ice thickness. We clarified this in the text in lines 711-716:

*"We therefore conducted some additional tests with both (THIZ and THIL) individual ice thickness datasets and for different combinations. For Pers glacier, the Langhammer dataset proved to be slightly better, while for Morteratsch, the Zekollari dataset performed better. But in general, all combinations tested gave more poor results, which encouraged us to use the average ice thickness in this study. It should be mentioned that the above analysis is valid for the glacier wide mean SMB deviations. For local point measurements, the uncertainty related to the used data, or the F-parameter, might be larger."*

**[RC1.GC5]** The justification of an exponential decay filter is not particularly strong, and I am concerned that the optimal distance results may be sensitive to the type of filter used. It would be worthwhile to consider other approaches or justify this choice (and its implicit assumption).

During the study, we conducted several tests with a mean, median, Gaussian and exponential filter. All with a constant size/variance/length scale over the study area and with a variable value for these parameters (depending on local ice thickness, local surface velocity etc). The exponential filter came out as the one with the best results for the smallest distance over which to filter, closely followed by the Gaussian filter. This also means that the ice flux divergence field which is

obtained after exponential filtering still contains the most detail, unlike filtering over very large distances which smears out the entire flux field. We agree with the reviewers that an exponential filter gives the local grid cell the most weight, but with a large length scale (which proved to be necessary – several times the local ice thickness), the surrounding cells also get a significant contribution. In addition, the tail of an exponential filter decreases more slowly towards 0, so that grid cells further away from the grid cell under consideration retain more weight than with a Gaussian filter.

A box method such as a mean or median filter gives equal weight to cell irrespective of their distance from the point in consideration. The issue we observed with these filters is that the effects of local phenomena (such as an ice fall or such as an acceleration in the glacier flow) are spread out uniformly when they are applied over larger distances which would imply that there would be no localized peaks in the flux divergence field. An exponential decay filter preserves this variation while filtering enough to get a smooth solution. Indeed, the local grid cell has the largest effect, but its value is "muted" by the spatial averaging. Our tests with the Gaussian filter showed only slightly less good results for the same/somewhat larger spatial scales. A Gaussian filter could therefore be an alternative, albeit with slightly less good results for this study.

To make this clearer in the text, we added some sentences to clarify our choice for the reader in lines 375-377

*"In this way, the effects of local phenomena (such as an ice fall or an acceleration in the glacier flow) are spread out uniformly when these filters are applied over larger distances, which implies that there are no localized peaks in the flux divergence field."*
and 758-760:

*"Other filters such as a mean filter or a Gaussian filter were explored, but all gave (slightly) less accurate results. However, our tests with a Gaussian filter showed only slightly larger errors for similar spatial scales. A Gaussian filter could therefore be a possible alternative to the exponential filter."*

**Minor comments:**

**[RC1.1]** L26-L27. Suggest '…several times the local ice thickness, accomplished in this study using an exponential decay filter.' There are several ways to consider the larger scale variations in stresses; your study nicely points to the exponential decay filter as one good approach.

We updated the text in the manuscript.

**[RC1.2]** L42. I'd suggest spelling out interpolation here and in the conclusions. Not so many letters but improves readability.

Done.

[RC1.3] L51-53. Suggest citing Berthier et al (2012) or similar studies here. Also a key point is that even if one corrects for ice flow, assumptions regarding ice density are needed to assess glacier mass balance, which are not the same assumptions as for the glacier scale (e.g. Huss, 2013).

Thank you for suggesting this paper. We included this reference.

[RC1.4] L60. There are many other relevant references here, and you can't be exhaustive, but see also Brun et al., (2018) and Wagnon et al (2020).

We thank the reviewer for suggesting this literature, which we added.

[RC1.5] L64. I'd be more explicit that many studies reduce the problem by using flux gates. I suppose this is a one-dimensional reduction and indeed smoothing, but it's also an entirely different kind of data processing.

We agree with the reviewer to make this more explicit and added a sentence to this section.

[RC1.6] L152. Is this the difference in SMB values or the rate?

This difference corresponds to the difference in rate. We added this to the manuscript.

[RC1.7] L171-183. The disagreement of the two thickness datasets is interesting and not terribly surprising. However, I am surprised that the authors have relied on the two modelled datasets (based on distinct field measurements and necessarily extrapolated using physical principles) rather than integrating the underlying field measurements of bed elevations in an updated ice thickness estimate. Surely this would result in a better consensus than the mean of two model results? This would allow you to also integrate your 2020 measurement(s).

See our answer in RC1.GC4.

[RC1.8] L188. These values differ by 18-24%

Modified.

**[RC1.9]** L255. Did you look at your off-glacier dH measurements (bias and standard deviation) at all? As with velocity, these provide an important measurement of error.

See our answer in RC1.GC4

**[RC1.10]** L260. Yes, these undulations are quite annoying from a geodetic perspective (and they mostly cancel one another out) but I find it strange to filter them out, when you could instead flow-correct the DEM (or underlying point cloud) using your velocity data (see Brun et al, 2018). An important test for this filtering is whether volume change is conserved before and after – do the volume-change measurements before and after the filtering equate? They are likely to be very close, but bear in mind that you are eliminating real volume changes; this is a good reason to consider warping your DEM to correct for glacier flow.

See our answer in RC1.GC3

**[RC1.11]** L277. Why is 25 in the denominator here? Should this be delta-x?

Corrected.

**[RC1.12]** L294. Please indicate what ImGRAFT settings you used, which correlator, etc, just for replicability

We have added this information in the text in lines 299-301:

*"Specific settings for the template matching of ImGRAFT are the use of the orientation correlation (OC) method and a large search height and search width of 200 m, to guarantee that all possible areas are covered."*

**[RC1.13]** L303. Are the glacier outlines from a standard source (which?) or did you digitize them yourself?

The glacier outlines were manually digitized based on drone and satellite images. We included a sentence to clarify this in lines 312-313:

*"The latter are manually digitized based on drone and satellite images and correspond to the end of the summer of every balance year under consideration."*

**[RC1.14]** L303. It's fine to remove them for your filtering, but the off-glacier velocity vectors give you an important metric of velocity uncertainty!

See our answer in RC1.GC4.

**[RC1.15]** L304. Do you median filter the u and v components or the speed (and direction)?

We filtered the u and v components and we added this in the text in lines 313-314:

*"velocity components that deviate too much from the surrounding grid cells"*

**[RC1.16]** L357. These studies don't really 'resample' to a lower resolution, they are instead resolving the flux across a gate or set of gates.

We agree and reformulated this in the text and we added a reference to Seroussi et al., 2011.

**[RC1.17]** L367. The choice of an exponential decay filter is interesting! I'm not fully convinced that an exponential is the best choice but neither is it a bad filter – have you tested or considered other filters for this? Unlike others (Gaussian, Weibull, mean/median), the exponential decay filter considers the closest observations to be the best estimate, giving the local value the highest weight. Do you have a reason to assume this to be the case? For instance, a block mean/median (as used for the dh/dt) would consider all nearby measurements to be a reasonable estimate. I see the exponential decay for a perturbation of ice thickness, but this is really the inverse process, no? I mean, the noise in the flux divergence map (which should be shown somewhere, by the way!) is in part due to ice thickness/velocity errors and in part due to the far-field longitudinal stresses. Did you test or consider other filtering approaches? For instance a Gaussian filter exhibits a similar distance decay, but considers any of the nearby estimates to be reasonable and weights them more or less equally.
L400. A concern here is that this filtering can internally 'break' conservation of mass. That is to say that the net flux divergence over your survey zone should equate to the flux into your survey zone; this should be the case with your gridded flux divergences, but after this filtering (which is not mass-conserving!) the net flux divergence (the integral of all pixel flux divergences) can change. Have you checked this? It is likely that the difference is within the uncertainty bounds of the flux into your lower domain, but this is a key problem with applying such filtering, and should be mentioned.

We thank the reviewer for this profound comment. Please see our answer in RC1.GC5.

**[RC1.18]** L403-407. The justification for applying both filters is not entirely clear to me from the text here. I see (later) that it does appear to improve the MAE results, but the text here could make the case more clearly for this.

Instead of applying one filter over a very long length, we found that applying two filters in a row over a smaller length improved the results and preserved more local phenomena. Furthermore, the filtering of gradients and/or the ice flux divergence for the glacier complex did not appear to be unconditionally the most optimal but the combination appeared to be. We clarified this in the text in lines 422-424:

*"This tends to reduce the distance over which the filtering must be performed, which is favourable to preserve local variations."*

**[RC1.19]** L446. Looks like the comma wasn't meant to be here

Removed.

**[RC1.20]** L451. The exception of the debris-melt-reduction is the terminal ice cliff, as identified by e.g. Immerzeel et al (2014).

Thank you for this suggestion. We added a sentence with a reference to Immerzeel et al., (2014).

**[RC1.21]** L453-456. It is certainly possible that this pattern is due to avalanches, but it's worth nothing as well that this is probably the least-constrained part of your survey area; do you have GCPs or GVPs that high?

During fieldwork, we were able to place and measure GVPs and GCPs in this region. In doing so, we immediately noticed the presence of avalanche remnants. We are therefore strongly persuaded to attribute these positive elevation differences to avalanche nourishment. We added a map of the location of the GCPs to support this.

**[RC1.22]** L467. 'are occurring' -> 'are evident'

Agreed and replaced.

**[RC1.23]** L490. 'Constant box filter' -> not clear what you mean here. Do you mean the average value of a box (e.g. altitudinal or longitudinal bin)?

We are referring to a box average filter that has a constant size over the entire glacier surface. In this case it is not, as we vary the size according to the local velocity. We are therefore taking flow variations into account. Because we decided to flow correct the DEMs, this comment is no longer applicable. See our answer in RC1.GC3.

**[RC1.24]** L499. What is the physical meaning of k=3? Three times the local surface velocity? The spatial pattern looks nice, but in my mind it would be much better to flow-correct your DEM as in Brun et al (2018). This of course represents a shift from an Eulerian to Lagrangian frame.

We agree with the reviewer to flow correct the DEMs and calculate the surface elevation changes instead of filtering this data. See our answer in RC1.GC3.

**[RC1.25]** Figures 5 and 7. dh/dt has units of m per year.

Added.

**[RC1.26]** Figure 8. The black dashed line for VM is missing on the right panel. These would represent the ideal cross section to assess whether your filtering has conserved mass (flux through cross section = integral of flux divergence).

We thank the reviewer for pointing this out. We have added the line to the right plot.

**[RC1.27]** L540-542. I'd avoid the speculation about a slowdown; you would need very different data and methods to assess this meaningfully!

Removed.

**[RC1.28]** L553. Section reference to 4.3 within 4.3

Removed and replaced by 3.4.

**[RC1.29]** L605. This shows a problem of the filtering used here – the possible local ablation increase, if present in the dH/dt, has been filtered out.

We removed this section since in our new version we do include point M26 in the analysis.

**[RC1.30]** Figure 12. Very nice summary; the processing has produced some very nice results in terms of reproducing the stake measurements!

Indeed! The method works pretty well. We look forward to see the application of our method on other glaciers where similar data are available.

**[RC1.31]** L627. Nice to see this lateral heterogeneity highlighted, and very nice to see the local deviations from a linear ablation gradient. Can you indicate the standard deviation of SMB for elevation bins (for example) based on your data?

This is an analysis that we considered when writing the manuscript, but which we decided not to include in the text.

**[RC1.32]** L638. See e.g. van Woerkom et al, 2019.

Thank you for suggesting this reference. We included it.

**[RC1.33]** L648-655. This is a really nice result, because it highlights an area where the assumption $b=b\_s$ breaks down. Do you have any measurements to isolate the surface and subsurface melt in these areas?

Unfortunately, we have no direct measurements in this area. During the fieldwork, we only observed large amounts of flowing meltwater at the bottom of these zones and a radial pattern of collapsing glacier walls.

**[RC1.34]** L667. I suggest making a note that the perturbation details will follow

Done

**[RC1.35]** L670. Is this MAE computed for all years' data or a single year?

The MAE is computed for all years. We clarified this in the text.

**[RC1.36]** L672-674. I'd appreciate a few more details about the implementation of these random perturbations. I can't tell if this is the equivalent of 'random Gaussian fields' or something else. Are the perturbation magnitudes also random (per realization? Per lump?).

Our perturbations are quite similar to random Gaussian fields but we deviate from them to the extent that the diameters of the patches are uniform for each perturbation. The perturbation magnitude is random per patch (=lump). We reformulated this in the text to clarify it in lines 684-686:

*"For the analysis, the different input fields are perturbed using random patches with a diameter (uniformly distributed) between 50 m (local errors at grid level) and 500 m with perturbation values normally distributed with a standard deviation of half the error estimate."*

**[RC1.37]** L679. Do the perturbations of F, thickness, and velocity also follow the same 'patch' approach?

Indeed, all analyses are performed in the same way, using the patches. We reformulated this in the text to clarify this.

**[RC1.38]** Figure 14. It is clear that the thickness perturbations have had the greatest effect, but these were also the greatest perturbation (30% vs 10% for velocity), due to the uncertainty of the underlying dataset(s). As such, these results do not really correspond to 'sensitivity' but somewhat more the 'uncertainty' attributable to an individual dataset. I.e. if the thickness data had a 10% uncertainty, how would its associated errors compare to the velocity-associated errors?

The reviewer makes a good point here. With a 10% uncertainty, the contribution of ice thickness and speed would be about the same. However, what we wanted to show here is that in the derived field (and in the way we have done it) the ice thickness and associated uncertainty of that ice thickness influences the result mostly. Following the reviewer's suggestion, we reformulated the title of this section into "uncertainty analysis".

**[RC1.39]** L705. My only criticism here is that only the exponential decay has been tested, and I am not sure that it is necessarily better or conclusive, so I would suggest softening the language around 'necessary' to instead indicate that the exponential filter produces closer correspondence to stake measurements.

Although several filters have been tested, we agree with the suggestion to mention that our research showed that exponential filtering is a good choice, but it is not a necessity as other filters can likely have similar results. We clarified this in the text and refer also to our answer in RC1.GC5.

**[RC1.40]** L738. I'm missing a discussion section considering implications/recommendations/future work? Density considerations? How important are the lateral variations in SMB (how much impact do they have on mass loss overall?) etc.

This was also suggested by the other reviewers. In the revised manuscript, we added a discussion section (section 6).

[revised manuscript text omitted]

---

## Author Comment (AC2)

In this document, we respond to the comments of reviewer 2 one by one. We also refer to the other documents with comments of the other reviewers to prevent repetition. Whenever some entirely new text has been added to the manuscript, it has been added in italics and in red below.

In the proposed revised manuscript, which is added at the bottom of this document, the textual changes have been added in red.

**Reviewer 2:**

The authors of this manuscript investigate the possibilities of combining observations of high-resolution surface elevation changes with information about ice thickness and flux-divergence for estimating local surface mass balance. Due to the time and cost intensive stake measurements for determining the glaciological mass balance, such an approach might be useful to extend the number of observed glaciers, at least for mass balance estimates across the ablation zone. In addition, such a method might provide much improved information about the spatial distribution of surface mass balance, which cannot be gained by the usual interpolation of stake measurements.

The paper is nicely written, the methods are clearly described and the data are well presented. Also the results are rather promising in relation to the potential of this method. During recent years there have been several groups working on similar approaches, which was probably timely, because of the availability of low-cost aerial surveys by the application of UAVs. Besides some minor issues, I only have a few more in-depth remarks, which should be considered within a revised version of the manuscript.

We would like to thank the reviewer for the various constructive comments that greatly improved the quality of the manuscript. We have thoroughly analysed the raised issues and formulated a response on that basis. We revised the manuscript accordingly.

**General comments:**

**[RC2.GC1]** The main problem I see in the workflow presented, is the use of the ice thickness data. I cannot see any reason why the mean of the two existing data sets will be the best solution. Rather, I would have expected that experiments are carried out for both data sets and based on the results, there is a discussion about the suitability of the existing data.

We agree that this may seem unclear at first glance. However, we have carried out extensive tests on the individual data sets. We refer to RC1.GC4 for our detailed response

**[RC2.GC2]** Also, I am not fully convinced about the application of the exponential decay filters for the flux data. Here, I would expect an improved argument, why such a filter should be used instead of others.

We have tested various filters and consistently the exponential filter turned out to be the most successful. We refer you to the answer in RC1.GC5 for further explanation.

**[RC2.GC3]** The results of the "continuity equation method" are compared to local stake field measurements. This is a fair approach. However, I miss a discussion about the validity of the method with respect to the ice thickness data. As far as I can see, ice thickness data have been collected close to the stake network. Therefore, also the extrapolated and filtered ice thickness data are close to the real values in these locations, which very likely improves the validity of the local results. Further away from the stake locations, a comparison of both data sets becomes much more difficult, as the SMB distribution from field measurements might not resemble reality (as is postulated in the manuscript), but the derived SMB based on interpolated ice thicknesses might also not provide the same degree of accuracy, as close to the measured profiles. Based on the existing data there is not very much, which could be done, but for me it seems appropriate to include this issue at least in the discussion of the results.

The reviewer makes an interesting observation here. However, we would first like to mention that for both the THIL and THIZ datasets, ice thickness measurements were also carried out for large areas in the ablation area in areas where no SMB stakes are present. The measurements of the Langhammer dataset were carried out over the entire ablation area, as this was done using RES from a helicopter. Furthermore, all ice thickness measurements and stake measurements were performed completely independent. Notwithstanding, we agree with the reviewer to mention more explicitly in the manuscript that the accuracy analysis is valid for the stake locations in lines 760-762:

*"Furthermore, the stakes used for validation were not entirely homogeneously distributed throughout the study area. The accuracy analysis may therefore have been biased towards these locations, which may also have influenced the choice of the optimal filter (length)."*

**Minor comments:**

**[RC2.1]** 16: please better distinguish between glacier-wide SMB and local point SMB

"point" SMB was added.

**[RC2.2]** 20: The "continuity equation method" needs some introduction

We added a sentence to already introduce the concept of conservation of mass. Further explanations are given in the introduction section.

**[RC2.3]** 27: The application of the exponential decay filter is not a-priori required for obtaining a suitable accuracy. There might be also other filters, which are suitable for this.

Agreed and modified in the text as this point was also raised by reviewer 1.

**[RC2.4]** 37: "Local energy budget", temperature ist not the primary driver of melt

Agreed and modified.

**[RC2.5]** 42: Not at every glacier, the stake network consists of a "small" number of stakes.

Agreed and removed.

**[RC2.6]** Geodetic methods are used since many decades, not only "lately".

Agreed and replaced.

**[RC2.7]** 66: It is not the lack of satellite data, but the lack of high-resolution observations.

Clarified by adding the word "sufficiently"

**[RC2.8]** Fig. 1: The strange shape of the snout of Vadret da Morteratsch needs some explanation. It is due to the debris cover, but it might be a good idea to explain that at some stage.

We clarified this in the text in lines 100-104:

*"The peculiar shape of the front of the Vadret da Morteratsch is caused by a combination of a large amount of debris on the western side that kept the ice below more insulated during the past years, and shading effects from the surrounding mountains. Since the 2000s, the front of the glacier has retreated significantly, but a large area of stagnant ice remained protruded to the north on the western side of the valley (Figure 1)."*

**[RC2.9]** 126: Is it possible to provide a maximum estimate of the melt effect during this period?

We do not have exact numbers on this as we did not take measurements at the same place at the beginning and end of the fieldwork periods. Based on our estimates, it is at most up to 10 centimeters at the front. In addition, the time period between the point SMB measurements and the UAV campaigns is maximally 3 days as the fieldwork campaign was always split into a part on Pers glacier and a part on Morteratsch glacier. We added this to the manuscript for clarification in lines 137-139:

"…the difference between the SMB measurements and surface elevation changes caused by not surveying simultaneously is estimated to be at most up to 10 centimeters, which is within the uncertainty bounds of the measurements."

**[RC2.10]** Table 1: Flight altitude is a bit misleading, as it is height above ground.

Agreed and modified.

**[RC2.11]** 136: Which service of swipos? Is there a reference for that?

This concerns the Swipos-GIS/GEO service.

**[RC2.12]** 148: This is just an "impressive" number. It would be more reasonable to provide the average number of stakes and the number of observation years.

We added the average number of observations in each year.

**[RC2.13]** 152: Are the numbers the difference? Otherwise, they should be negative.

We agree with the reviewer to make these values negative as they indicate the difference in SMB rate.

**[RC2.14]** 179: GlaTe requires reference.

We added a reference to Langhammer et al., 2019

**[RC2.15]** 182: The reference year is 2001 according to Zekollari et al., 2013.

We are aware of this date in Zekollari et al., 2013. However, it turned out that the DEM which is used is valid for 1991.

**[RC2.16]** 187: Are these values corrected to a common year? Please provide this information. Also, Zekollari provides an accuracy estimate of 50 m for the maximum ice depth.

These values were indeed corrected to a common year (2020). We added this in the text. The uncertainty of 50 metres is discussed in lines 187-188. We note that the difference still falls between the mentioned error bounds.

**[RC2.17]** 192: This is not a hypothesis, it is a fact that the two distributions will provide different results.

This was modified.

**[RC2.18]** 194: "thickest point", rather "thickest region", as you cannot be sure that the absolute maximum was covered by the measurements.

Agreed and modified.

**[RC2.19]** 203: Your minimum assumption is for regions with a today ice cover?

Correct. To further clarify this for the reader, we have reformulated this in the text

**[RC2.20]** 212: "bedrock elevation inferred", I guess these are the areas, where the h_min of 5 m is applied?

This is correct.

**[RC2.21]** 232: "because", typo

Rectified.

**[RC2.22]** 260/292: Why do you filter these patterns? You could use the surface velocity field to correct for them.

We refer to our answer in RC1.GC3.

**[RC2.23]** 303: There is no mentioning of using the velocities outside the glacier for correction/quality control.

As this issue was also raised by Reviewer 1, we refer to our answer in RC1.GC4 for this comment.

**[RC2.24]** 388/389: Did you take into account the uncertainties of the input parameters? Ice thickness has large errors, which increase with thickness.

Indirectly, this is accounted for by increased filtering of the data in areas of thicker ice. In addition, in the uncertainty analysis (section 5.1) we use an uncertainty percentage which takes into account errors that are larger for larger ice thicknesses.

**[RC2.25]** 413: Where does the uncertainty of the surface SMB come from? There might be quite some variation in the perimeter of 25 m, if the surface is not homogeneous.

This is an estimate of the uncertainty of the measurement itself based on terrain analysis during the fieldwork. Values between 0.1 and 0.3 m are often used for ablation areas.

**[RC2.26]** 455: Is this region part of the SMB considerations? If yes, is a density correction included?

This part is not part of the study area for the SMB (see Figure 10). However, we agree with the reviewer about the important issue of density corrections and added a section in the discussion where we further elaborate on this.

**[RC2.27]** 470: standard deviation of what?

This concerns standard deviation of the elevation changes. But since this section has been removed, it is no longer included in the manuscript.

**[RC2.28]** 515/516: What does "deformations of the surface" mean in this context?

In certain areas, the surface has changed to such an extent that it cannot be traced between two consecutive years. An example are the areas with a complex pattern of crevasses or the repositioning of supraglacial melt streams between two years.

**[RC2.29]** 564: Should the value be negative?

Thank you for pointing this out. We replaced 4 with -4.

**[RC2.30]** 569: What about the THIL dataset? Do you rely only on the THIZ data set in this region?

In this region, as in all others, we use the average of the two data sets. We reformulated this in the text to clarify it.

**[RC2.31]** 589: -13 m/yr: I cannot see any values close to this number in Fig. 12.

We believe that this is figure 10 in the (revised) manuscript. Close to the front of Vadret da Morteratsch, values are between -12 and -13 m.i.e yr$^{-1}$. We agree with the reviewer that this is not clear and therefore, we changed the interval of the isolines.

**[RC2.32]** 629/630: This is an area, where the accuracy of the ice thickness is probably lowest due to the lack of measurements and problems with the numerical representation.

We agree with the reviewer that the ice thickness in this area is the least constrained by measurements. However, with a very low flow velocity and a small ice thickness, the ice flux divergence values in these parts are very small in any case and the SMB is almost completely in line with surface elevation changes. We therefore do have confidence in the reconstructed SMB-value in these areas.

**[RC2.33]** 634: The "-12 m/yr" are a result from the continuity equation. Such an ablation rate is rather unlikely, unless there is a very thin layer of debris, which enhances melt.

Thank you for mentioning this. It is indeed true that there is a thin layer of debris in this zone at the terminal ice cliff. We have added this in the text.

**[RC2.34]** 635: This needs some supporting information, as debris cover is only in rather special cases related to the activity of lateral moraines.

We added a sentence in which we indicate that we regularly observed boulder supply towards the glacier. In addition, our surface elevation change maps indicated a loss of sediment towards the glacier surface in lines 649-651:

*"Such thicknesses are very likely in this area. During the fieldwork campaigns, boulder supply from the moraines was observed on several occasions, which was also documented in the Himalaya by Van Woerkom et al. (2019)."*

**[RC2.34]** 693/694: This relates to glacier wide mean values. The sensitivity is probably much larger for local difference between measured and modelled values.

We agree with the reviewer and specifically mention in the revised version that from local point measurements, the uncertainty related to the use of the data, or the F-parameter, might be larger.

**[RC2.35]** 756/757: SMBs are compared at locations where ice thickness information is mostly based on measurements (the thickness profiles are along the stakes at least for THIZ). Therefore, it is difficult to assess the accuracy for regions further away from the measurements.

We agree to explicitly mention this in the text. We refer to our answer in RC2.GC3.

[revised manuscript text omitted]

---

## Author Comment (AC3)

In this document, we respond to the comments of reviewer 3 one by one. We also refer to the other documents with comments of the other reviewers to prevent repetition. Whenever some entirely new text has been added to the manuscript, it has been added in italics and in red below.

In the proposed revised manuscript, which is added at the bottom of this document, the textual changes have been added in red.

**Reviewer 3 – Alexander Raphael Groos**

This is an interactive comment on the manuscript by Lander Van Tricht et al. (hereafter the authors) entitled "Estimating surface mass balance patterns from UAV measurements on the ablation area of the Morteratsch-Pers glacier complex (Switzerland)". In their study, the authors investigate the potential of surface elevation change and surface velocity data obtained from repeated UAV surveys in combination with ice thickness data to produce high-resolution ice flux divergence and surface mass balance maps of the ablation zone of alpine valley glaciers. This contribution is very welcome as it introduces an approach to determine spatial variations of the glacier mass balance, which can hardly be assessed by stake measurement alone. In view of the ever-increasing number of high-resolution topographic data, UAV-based surface mass balance investigations seem to be a viable complementation to glaciological mass balance observations. Moreover, the described method might also be useful to investigate the surface mass balance of glaciers that are difficult to access. Whether the UAV-based approach (especially when relying on GCPs rather than on RTK) is less time-consuming than classical ablation stake measurements is, however, questionable. The manuscript is well-structured, and the methods are clearly described. However, I have some remarks and questions, mainly concerning the presented UAV data. I have tried not to repeat the reviewers' comments, but there may still be some overlaps.

We thank Alexander Raphael Groos for his positive appreciation of the paper. We have addressed the questions and comments below and updated the manuscript where needed. Our revised version now contains more detailed information regarding the UAV surveys and data processing.

**[CC1.1]** Did you experience any difficulties or problems during the areal surveys that should be considered by other groups when applying this method in the future?

We have been doing the same surveys for several years (actually since 2014) and have found a clear workflow. It is of course crucial to cover the whole area under observation with the UAV surveys. Morteratsch-Pers glacier is quite large and complex, which requires us to split the flight plan into several parts. It is thereby crucial to let the areas overlap so that they can be easily attached to each other afterwards. We also provided always GCPs in the overlapping area in order to georeference two connected flight areas with the same points.

We clarified this in the text in lines 134-135:

*"We ensured in every case sufficient overlap between the different flight plans so that they could be easily attached to each other afterwards"*

and in lines 155-156:

*"Specific attention was paid to the distribution of GCPs in the overlapping areas of the different flight areas in order to be able to georeference two neighbouring flight areas with the same points."*

**[CC1.2]** On which days were the aerial surveys performed. Did the illumination conditions (e.g. cloud cover) change during the 4-6 days field work period and did this affect the image processing?

We performed our flights always between 25 September and 2 October. During most of the flights the weather was sunny. During cloudy moments, we always checked if there was enough contrast, but this was always the case. We never adjusted our image processing and apparently, this did never influence the amount of keypoints detected on the images in Pix4D.

**[CC1.3]** Can you estimate the melt rate and surface lowering during these days? If 4 to 6 days passed between the aerial surveys and the melt rate was in the order of 3-4 cm day$^{-1}$, this would translate into surface an elevation change in the order of 12 to 24 cm (if ice flow is ignored). Did this affect the image processing and the generated digital surface models in any way?

We refer to our answer in RC2.9 for more information about this.

**[CC1.4]** How were the GCPs distributed across the ablation area? Could you include the position of the GCPs in one of the overview maps (e.g. Fig. 2), at least exemplarily for one year?

Done. We added the GCPs for 2019 in Fig. 2

**[CC1.5]** How many of the GCPs were used for "calibration" and how many for "validation" (GVPs)? Better distinguish between GCPs and GVPs from the beginning.

Done. We reformulated some sentences in section 2.1 to already introduce the concept of GVPs in this part. Typically, we used about 20% of our GCPs as a GVP.

**[CC1.6]** I think the major drawback is that the "stable terrain" outside the glacier area was not considered to assess the accuracy of the DMSs. You mentioned that the vertical accuracy of the Trimble 7 GeoXH RTK GPS used to measure the GCPs is in the order of 20-30 cm. As stated in Table 2, the mean absolute error (MAE) of each DSM is less than 10 cm. This means that the DSM are self-consistent and very accurate (at least relative to the considered GCPs), but the MAE does not

tell you anything about the xyz-offset between the DSMs of the different years. Therefore, I would suggest to compare the DSMs over stable terrain (in case you covered such an area during your surveys).

We thank the reviewer for these suggestions which are much appreciated. We added a section in 4.1 in which we use stable terrain for an error analysis concerning the elevation changes. See our answer in RC1.G4.

**[CC1.7]** In 2020, you used a UAV with RTK. I assume that in this case you considered the distributed GCPs only for validation. Is this correct?

We did use some GCPs for the 2020 surveys. This would improve the accuracy of the DSMs. Nevertheless, we distributed less GCPs (see Table 2).

**[CC1.8]** Abstract: The acronym SMB is defined in the abstract, but the acronym UAV not. Define both in the abstract or introduction.

Done.

**[CC1.9]** L68: Maybe use a gender-neutral term such as "Unoccupied Aerial Vehicle (UAV)" that have been introduced recently (Joyce et al. 2021) and become more and more popular in the community.

Done. We used "Unoccupied Aerial Vehicle".

**[CC1.10]** L69-72: Maybe mention here that repeated UAV surveys have been conducted before by other groups in different regions to derive surface velocities (e.g. Kraaijenbrink et al., 2016; Benoit et al., 2019) or to compare surface elevation change with ablation stake measurements (e.g. Groos et al., 2019*), emphasising the need for or potential of a transferable method to use such topographic data to determine the SMB distribution.

Thank you for suggesting this. We included it in the introduction.

**[CC1.11]** Figure 1: It's a personal preference, but I think for international readers geographic coordinates (LatLon) would be more informative.

We understand the comment, but we prefer the use of the local date as it allows intuitive estimation of distances.

**[CC1.12]** L117-119: Where no surveys performed in 2017 or was the photogrammetric processing not successful? Other studies have shown that the SFM-technique in principle also works for snow-covered areas (e.g. Bühler et al., 2016). Or was it impossible to distribute GCPs under this circumstances?

It was a combination of both. Our method of spreading GCPs consists of covering the area of the flight plan with GCPs as best as possible, then performing the flight, then measuring and removing the GCPs. During the scheduled days in summer 2017, foggy conditions and snowfall prevented us from monitoring the upper part of Pers glacier. The periods during which we could have flown were too short and spreading GCPs was not possible, so we chose to perform only the SMB measurements on the stakes. We did try it for a while, but due to our relatively high flight altitude above the surface, low clouds occurred between the surface and the drone several times during the test flight.

**[CC1.13]** L126: Can you estimate the uncertainty? See general comment.

We refer to our answer in RC2.9. Concerning surface melt, we expect a maximum of up to 10 centimeters at the front of Vadret da Morteratsch, which is within the uncertainty bounds of the measurements.

**[CC1.14]** Table 1: Could you provide some more information here: e.g. flight dates, range of height above ground level, no. of images acquired, size of surveyed area...

We added some additional details (especially for 2020) in this table. As far as the exact dates are concerned, it seems for us somewhat too detailed to mention them in this table for all 4 years.

**[CC1.15]** L134: Can you include the position of the GCPs (at least exemplarily for one year) in one of the overview maps, in Fig. 2?

Done. We added the GCPs measured in 2019 in Figure 2.

**[CC1**.16] L141: In case of the 2020 surveys, were the GCPs only used for validation? Is it realistic that the P4RTK system is more accurate than the Trimble 7 GeoXH RTK GPS? Any comparative tests on stable terrain?

See our answer in CC1.7. We did not perform some specific comparative tests on stable terrain. However, because we used some GCPs (measured with the Trimble GPS, we are confident that the accuracy is similar.

**[CC1.17]** L148: Can you provide the total number of ablation stakes rather than the number of measurements?

Done. We provided the average number of stakes in each year (~ 15).

**[CC1.18]** Figure 3: The two reviewers already commented on that. Why is the average of both datasets the best choice? Would there be any arguments for using one over the other. Anyway, the sensitivity analysis is appreciated. Would it be possible to include the conducted radar measurements pathways in panel a and panel b? It would be helpful to use the empty lower right panel to include a difference map of THIZ and THIL to highlight areas of good agreement and areas with larger uncertainties.

We refer to our answer in RC1.GC4. Further, we added a panel in Figure 3 to show the difference.

**[CC1.19]** L248: Why did you choose the old Swiss Grid (CH1903 LV03) rather than the new one (CH1903+ LV95)?

We updated our figures to the new Swiss Grid.

**[CC1.20]** L250: How many of the GCPs were used as GVPs? It would be fair to provide some more details and, if possible, indicate them in one of the maps (e.g. in Fig. 2).

Done. See our answers in CC1.5 and in CC1.15.

**[CC1.21]** L252: Sometimes you use 5-10 cm and sometimes 0.05-0.10 m. Try to be consistent.

Rectified.

**[CC1.22]** L431-432: The stated MAE defines the accuracy of a DSM relatively to the used GCPs, but it does not tell you anything about the "absolute" accuracy. This can only be assessed by considering data from "stable terrain" outside the glacierised area. The vertical accuracy of the Trimble 7 GeoXH RTK GPS was stated to be in the order of 20-30 cm, so it is likely that the difference between DSMs from different acquisition dates is larger than the stated MAE in Table 2.

See our answer in CC1.6

**[CC1.23]** Table 2: Does the GCP density also include the points used as GVPs?

It does. But as you can see on Figure 2, we used a very large amount of GCPs.

**[CC1.24]** L453-457: Did you place GCPs in the relatively steep area? If not, do you think the observed positive surface elevation changes between 2019 and 2020 could be the result of inaccuracies of the DSMs (especially at the margin of your study area) rather than a mass gain related to increased avalanche activity? It's not necessarily the case here, but DSMs are prone to large-scale distortions (e.g. warping) if no GCPs are distributed at the margin of the study area (e.g. James and Robson, 2014; Groos et al., 2019*).

This is indeed an issue which we observed during our first campaigns in 2014-2016. However, during the 2017-2019 field campaigns, we always ensured a very large number of GCPs that were distributed as homogeneously as possible, also at the edge of flight plans. The highest part of the ablation area of Pers glacier, where we observed the positive dhdt, is relatively flat, which allowed us to maintain the same GCP density in this region (see Fig. 2). In addition, we observed old avalanches and firn fields during fieldwork, giving us confidence in the elevation differences presented in this region.

**[CC1.25]** I would suggest to include a discussion section to elaborate on the implications of your study and recommendations for future work. Regarding the transferability of the presented approach, it would be interesting to discuss the uncertainties related to the use of modelled ice thickness data (e.g. Farinotti et al., 2021) when applying your method to determine spatial SMB variations of glaciers in data-scarce regions. Are there any limitations or challenges that should be considered when applying this method to mountain glaciers with a different setting (e.g. varying geometry, varying surface velocities, varying debris cover extent, presence of ponds and ice cliffs). Moreover, glaciers, for which multiannual high-resolution topographic data from repeated UAV surveys already exist (e.g. Kraaijenbrink et al., 2016; Benoit et al., 2019 Groos et al., 2019*), could be briefly mentioned as potential sites for the further testing of the presented method. Recommendations regarding best practices for the implementation of UAV-surveys in mountainous terrain could also be included here.

We thank the reviewer for giving these suggestions. The other reviewers also mentioned this. We added a discussion section in which we elaborated among others on the suggested topics.

[revised manuscript text omitted]

---

## Author Response (AR2)

In this document, we respond to the final comments and suggestions of the reviewer Evan Miles one by one. The new text that has been added to the manuscript, is added in italics and in red below.

> I'd like to congratulate the authors on a nicely structured and well-presented study, which was a pleasure to read. In their manuscript revision, the authors have carefully addressed all my comments (and those of the other reviewers). They have consequently made careful adjustments to their methods and throughout the manuscript text. I have a few minor comments below that have no impact on the calculations, discussion, or conclusions. I am very happy to recommend the manuscript for publication following these technical corrections.

We would like to thank reviewer Evan Miles for his (final) comments and suggestions which were really appreciated.

**Specific comments:**

> **[RC1.1]** L52. Suggest 'do therefore not' -> 'therefore do not'

We updated the text in the manuscript.

> **[RC1.2]** L96. Suggest 'both glaciers disconnected' to 'the glaciers disconnected' as the glaciers are disconnecting from one another rather than from a third body

Done

> **[RC1.3]** L112-113. A bit awkward – '...has been measured twice [], and again in 2020...' perhaps simplify to '... has previously been measured twice [].' as the new measurements are described in

Done

> **[RC1.4]** L208-214. I'd recommend some basic depiction of the new measurements in an Appendix (Table?) or Supplementary Material, or you could indicate transects or spot measurement locations in Figure 3c. Perhaps mention these data in the Code and Data Availability statement -> I guess they have been or will be submitted to GlaThiDa?

We added the transect in Figure 3c. The data might be submitted eventually to the GlaThiDa.

**[RC1.5]** L209. Suggest 'considerable effect for' -> 'considerable effect on'

We changed this in the text.

**[RC1.6]** L290. A small detail, but Brun et al (2018) tested different sizes of Gaussian filter for this calculation. What size Gaussian do you use?

We added this information in the manuscript:

*"with a 25-pixel kernel size (50 m)"*

**[RC1.7]** L408. The ratio correction works since the surveyed area is almost entirely within the ablation area. However, if your survey area was the entire glacier, the net flux divergence would be zero by definition, and the ratio approach would set flux divergence to zero everywhere – not ideal! I don't think you need to recalculate with a different method, though; instead, perhaps just remind the reader of the survey domain 'To provide for conservation of mass _within our survey area_'

Thank you for this useful remark. We added this in the manuscript.

**[RC1.8]** Figure 6 is referenced before figure 5.

We decided to keep the current order because it follows the order of the text and results more.

**[RC1.9]** Table 2. Could you additionally indicate the number of GVPs?

Done

**[RC1.10]** L473. It looks like the coregistration of the 2017-2018 DEM pair is slightly inferior to the other pairs (looks like the 2017 DEM might be shifted). I don't think this is a problem but it's worth keeping in mind in the discussion and your uncertainty testing – this was your first UAV survey in the area and had the least constraint from control points, so this is not surprising. My main question from this is – what level of uncertainty to ascribe to the DEM-differencing? To me it looks like 20cm is ample for 2018-2019 and 2019-2020, while a higher value is needed for 2017-2018. It is not surprising that this is slightly higher than you might derive from Table 2, but these domains

are to some degree the worst-case for repeat surveys (steep slopes), entirely independent from the GCPs, and this metric includes the uncertainty from each of 2 surveys. The only change to make is that in the uncertainty section you should make reference to these results to justify the chosen perturbation for dh/dt

We added a reference to Table 2.

**[RC1.11]** Figure 6. Nice depiction of the results before and after flow-correction.

Thank you

**[RC1.12]** L515 and 516. Possibly broken internal cross-reference links to Figure 7?

Thank you for remarking this. We added the reference in the manuscript.

**[RC1.13]** L521-525 could go into methods after current L320

We agree that this part could be replaced to the method section, but we decided to keep the present structure, as it fits also with the current presentation of the results.

**[RC1.14]** L534. The contours are for velocity (m yr-1 rather than m), right?

Correct. We rectified this in the document.

**[RC1.15]** L618, 630, 631. Broken cross-reference to a Figure

We added the correct references.

**[RC1.16]** Figure 12. Nice! It would be very interesting to quantify the mean or median absolute deviation of differences, and possibly the mean error ablation error from a simple linear fit to stakes; these would be relatively simple numbers to put in the text ~L668

Thank you for this suggestion. We fully agree that this is very interesting and we added this in the manuscript:.

*"The mean of the differences corresponds to 0.56 m i.e. yr$^{-1}$, which is specifically caused by the western side of the Morteratsch glacier with a thick layer of debris (see Figure 2 and Figure 12). The simple linear fit would overestimate the ablation in the surveyed area with 2.4 \* 10$^6$ m$^3$."*

**[RC1.17]** L712. The qualitative description here is fine, but it would be nice to have a basic numerical justification to support the findings. Perhaps you could indicate a representative MAE for the THIZ/THIL setups here as for the discussion in 5.2.

Done

**[RC1.18]** L739. I'd recommend to give the actual value here (PM average).

Done

**[RC1.19]** L786-787. Can you provide the values here?

Done

*"Concerning Vadret Pers, the MAE and SEE were quite similar (0.52 and 0.62 m i.e. yr$^{-1}$ respectively) while for Vadret da Morteratsch, the MAE and SEE were considerably higher (1.08 and 1.22 m i.e. yr$^{-1}$ respectively)."*

**[RC1.20]** L789. The method can certainly be applied, but may not be as _robust_

We agree with the reviewer. We will test the method additionally on other glaciers. The method can also be tested on other glaciers where sufficient data is available (examples are mentioned in the manuscript)

---

## Author Response (AR3)

Final response letter to the editor Etienne Berthier.

Dear Authors,

Thanks a lot for revising carefully the manuscript and taking into account the minor or technical comments made by both referees.

I am delighted to accept your study for publication in TC.

I have only a very few minor suggestions below, the most important in my view being the added value of providing the mean error for the modelled SMB vs. in situ data.

Best regards,

Etienne Berthier

We would like to thank the editor for his contributions and suggestions to this manuscript. We have carefully reviewed the latest comments and implemented the suggestions.

**Specific comments:**

L30. "Hamper" is rather strong : Miles et al., 2021 showed recently that the method can be applied to satellite data for glaciers larger than 2 km² in HMA. But it is true that uncertain ice thicknesses increases uncertainties from these remote sensing estimates so maybe modify the text a little bit.

We replaced the word "hamper" with "complicate" which in our opinion has the correct meaning in this context.

L75. Are there others studies than Vincent et al., 2021 doing so? Otherwise use singular form and do not use "e.g." before the ref.

We are not aware of any other study and therefore reformulated this sentence.

L470. "in consideration" not needed

Done

| L489. "significantly" I think |
|---|
| Done |

| L495. parenthesis not closed |
|---|
| Solved. |

| L595. "elevation" seems better here |
|---|
| Replaced. |

| Fig 10. the -1 tick is shorter. |
|---|
| Adjusted. |

| L631. why not providing also the Mean Error (ME) to quantify if the SMB field has any bias? I really feel that it would be a useful information to confirm the almost unbiased nature of the UAV estimate. |
|---|
| Thank you for this suggestion. We added the ME which appeared to be indeed close to zero (as expected). |

| L699. "which…. Accuracy". You could simply write 'taken from Zekollari et al., (2013)' |
|---|
| Done |

| L712. It is not the surface velocity which has this SD, rather the SMB. Reword. |
|---|
| Done |

L787. A reference for the 0.5% value? Maybe the paper presenting the ice thickness database (GlaThiData)?

We added this reference.

L806. "centimeter accuracy" maybe true for dh, but unit of velocity is distance per time so the statement is not homogeneous.

Solved.

L812. maybe quote the 0.5 m i.e. /yr bias that you found if stakes only were used?

This value concerns the deviation between measured and modelled SMB. We decided to not mention this value her.

L868. This came as a surprise to me. Was this an internal review? Then it should be in the acknowledgment section, not here. (or indicate clearly this is an internal review).

Alexander Raphael Groos gave a public comment on the manuscript which we considered as a review. We removed this sentence and included it in the acknowledgment section.